# Extracting task-relevant preserved dynamics from contrastive aligned neural recordings

**Yiqi Jiang**[1]*, **Kaiwen Sheng**[1]*, **Yujia Gao**[2], **E. Kelly Buchanan**[1], **Yu Shikano**[1], **Tony Hyun Kim**[1,4], **Yixiu Zhao**[1], **Seung Je Woo**[1], **Fatih Dinc**[1,3], **Scott Linderman**[1], **Mark Schnitzer**[1,4]

[1]Stanford University     [2]Carnegie Mellon University
[3]University of California, Santa Barbara     [4]Howard Hughes Medical Institute

{yqjiang, kwsheng, swl1, mschnitz}@stanford.edu

## Abstract

Recent work indicates that low-dimensional dynamics of neural and behavioral data are often preserved across days and subjects. However, extracting these preserved dynamics remains challenging: high-dimensional neural population activity and the recorded neuron populations vary across recording sessions. While existing modeling tools can improve alignment between neural and behavioral data, they often operate on a per-subject basis or discretize behavior into categories, disrupting its natural continuity and failing to capture the underlying dynamics. We introduce Contrastive Aligned Neural DYnamics (CANDY), an end-to-end framework that aligns neural and behavioral data using rank-based contrastive learning, adapted for continuous behavioral variables, to project neural activity from different sessions onto a shared low-dimensional embedding space. CANDY fits a shared linear dynamical system to the aligned embeddings, enabling an interpretable model of the conserved temporal structure in the latent space. We validate CANDY on synthetic and real-world datasets spanning multiple species, behaviors, and recording modalities. Our results show that CANDY is able to learn aligned latent embeddings and preserved dynamics across neural recording sessions and subjects, and it achieves improved cross-session behavior decoding performance. We further show that the latent linear dynamical system generalizes to new sessions and subjects, achieving comparable or even superior behavior decoding performance to models trained from scratch. These advances enable robust cross-session behavioral decoding and offer a path towards identifying shared neural dynamics that underlie behavior across individuals and recording conditions. The code and two-photon imaging data of striatal neural activity that we acquired here are available at https://github.com/schnitzer-lab/CANDY-public.git.

## 1 Introduction

Understanding how complex behavior arises from high-dimensional neural activity is a central goal in systems neuroscience. Several studies have found that high-dimensional neural activity can often be reflect a lower-dimensional set of latent states that evolve over time as an animal perceives, plans, and executes actions [1–7]. Notably, growing evidence indicates that behaviorally relevant latent states and dynamics are preserved across recording sessions and subjects [8–10]. However, reliably uncovering these preserved latent variables is challenging due to substantial variability in neural recordings and behavior among animals. Specifically, different sessions often measure different subsets of neurons, and trial durations and timing can vary widely as animals initiate or complete movements at different speeds. These discrepancies make it difficult to consistently identify and align preserved latent dynamics across time and subjects.

---

*These authors contributed equally to this work.

39th Conference on Neural Information Processing Systems (NeurIPS 2025).

Traditionally, researchers have modeled each session separately and then aligned the latent spaces post hoc using techniques like canonical correlation analysis (CCA) [10–13]. This approach often assumes that trials have identical durations, which can limit its applicability to datasets with variable trial lengths. On the other hand, unsupervised approaches to modeling neural dynamics addresses the variability across sessions by employing session-specific encoders to project neural activity into a common latent space, but it does not explicitly ensure that the resulting latent dynamics are behaviorally aligned [14, 15]. More recently, CEBRA aligns embeddings across sessions based on behavioral similarity; however, it discretizes behavior into categories[1], which overlooks the natural continuity of behaviors, and it does not model the temporal dynamics of the latent space [16]. While state-space models, such as linear dynamical systems (LDS) [17–19], switching linear dynamical systems (SLDS) [20, 21], and nonlinear state-space systems [22, 23], model temporal structure, they lack built-in mechanisms to align dynamics across sessions.

In this study, we introduce CANDY, Contrastive Aligned Neural DYnamics, a novel framework that integrates neural recordings from multiple sessions and extracts preserved, behaviorally-relevant latent dynamics for a task. CANDY simultaneously aligns multi-session embeddings and estimates latent dynamics, eliminating the need for post-hoc alignment and enabling direct recovery of interpretable temporal structures across sessions. CANDY projects neural activity from different subjects onto a common space using a subject-specific encoder (Fig. 1**A**). It uses contrastive learning to align the latent embeddings, guided by the continuous behavioral data across sessions, such that the latent embeddings capture shared, behaviorally relevant factors (Fig. 1**B**). At the same time, CANDY learns latent dynamics with an LDS to predict how latent states evolve over time (Fig. 1**C**).

Our key contributions include: 1) Methodologically, we develop a new framework to extract preserved dynamics underlying continuous behavior across sessions and animals using a rank-based contrastive objective and a linear dynamical system. 2) Conceptually, we show that contrastive alignment is essential for uncovering preserved dynamical systems, not merely a decoding aid. 3) Biologically, we show that a dynamical model trained on one set of animals transfers to held-out individuals, indicating the existence of preserved dynamical structures in a specific brain region under a given task. 4) Experimentally, we collected a new two-photon calcium imaging dataset in mice striatum across multiple days and animals.

## 2  Related work

**Neural latent dynamics**    Several methods have been developed for extracting low-dimensional latent factors from high-dimensional neural measurements [24–43]. Many methods also model the temporal dynamics of the latent factors using, for example, state-space models [24–26], sequential variational autoencoders [14, 40], low-rank recurrent neural networks [41, 42], diffusion models [43, 44], or teacher-student distillation [45]. However, these models typically focus on uncovering the dynamics of a single recording session, instead of the dynamics that are preserved across multiple sessions.

**Multi-session stitching in neuroscience**    Early efforts to learn shared neural latent spaces across sessions and subjects often required post-hoc alignment [9–11]. More recent approaches have projected neural data from different sessions to a common space via session-specific MLPs without explicit alignment or incorporated behaviors as supervisory signals to constrain the latent space [14, 16, 46]. In parallel, there has been a trend in building large-scale transformer-based foundation models by integrating heterogeneous neural recordings from multiple experiments and labs [47–52]. However, both approaches leave unclear which parts of the common space are actually aligned across sessions and what their behavioral relevance is. Additionally, the common latent space is less interpretable without explicit characterization of its dynamics.

**Contrastive learning**    Contrastive learning has emerged as a powerful technique for representation learning across modalities [53–59]. Previously, it has achieved great success in aligning different discrete modalities across multiple domains [55–58]. Recently, machine learning scientists have developed regression-aware representation learning techniques by contrasting samples against each other based on their ranking in the target space [53, 59]. Such contrastive learning has been shown to be effective for extracting shared temporal structure from time-series data across modalities in other

---

[1]Though CEBRA allows continuous behavioral variables, it treats continuous labels as many discrete classes.

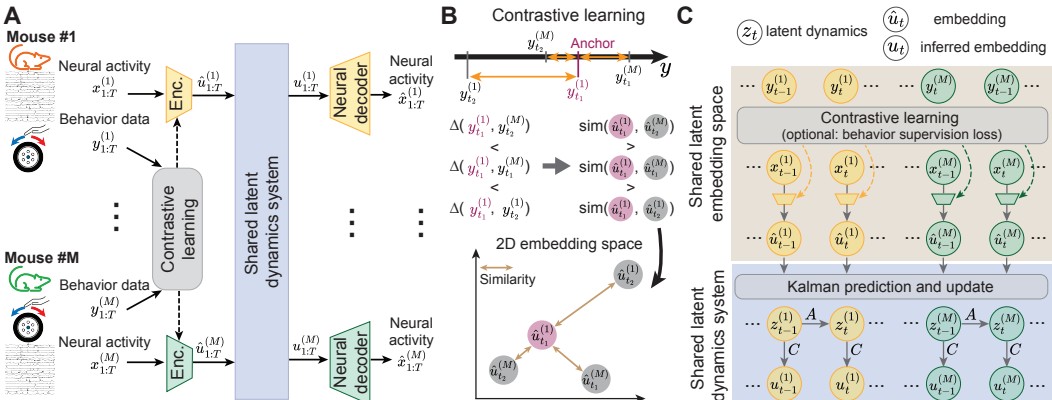

Figure 1: **Illustration of the CANDY architecture that aligns latent embeddings and extracts the preserved dynamics across sessions. A.** The model architecture includes aligning the neural embeddings across sessions and extracting a preserved task-relevant LDS. **B.** Illustration of the rank-based contrastive learning [53]. For an anchor embedding (colored in purple), other embeddings are ranked by their $L_1$ distance with respect to the anchor's behavior in the 1D behavior space. The loss then enforces that distances in the 2D embedding space reflect this behavioral ranking. **C.** A zoom-in view of the shared behavior constrained latent embedding space and shared latent dynamics system depicted in (**A**). The latent dynamics is modeled as an LDS with a dynamics matrix $A$ and an emission matrix $C$ to the latent embeddings space shared across sessions.

fields, such as videos, sensing physics, and robotics planning [60–62]. In neuroscience, recent studies have explored contrastive learning for extracting meaningful neural latent representations. CEBRA [16] embeds neural activity with noncausal convolutional neural network by using a contrastive objective on temporal similarity and discretized behavioral similarity. MARBLE [63] introduces a geometric deep learning framework that embeds local flow fields computed from neural activities into a latent space using contrastive objectives. However, it requires post-hoc alignment for downstream behavior decoding across sessions. While both methods produce useful low-dimensional representations, they do not explicitly model shared latent dynamics across sessions or individuals.

## 3 Methods

Extracting task-relevant neural dynamics that are preserved across recording sessions, spanning days and subjects, is essential to understand the common strategy that underlies behavior across days and subjects. In this section, we present CANDY, an end-to-end framework that combines behavior-anchored contrastive embedding alignment with a shared LDS. In Section 3.1 we describe our nonlinear encoder, which uses a behavior-anchored contrastive loss to align embeddings across all sessions; Section 3.2 details how we fit a shared LDS to these embeddings; and Section 3.3 presents the joint training strategy that integrates both objectives into a unified optimization.

Suppose we have $M$ sessions (*e.g.*, different days or animals), indexed by $s = 1, \ldots, M$. Session $s$ comprises $L^{(s)}$ trials, where trial $i$ has $T^{(s,i)}$ time bins. At each time bin $t$, we observe the high-dimensional neural activity $x_t^{(s,i)} \in \mathbb{R}^{N^{(s)}}$ and the associated continuous behavior $y_t^{(s,i)} \in \mathbb{R}^{n_y}$, where $N^{(s)}$ is the number of recorded neurons in session $s$ and $n_y$ is the dimensionality of the behavioral measurement. For notional simplicity, we drop the trial index $i$ in the following sections.

### 3.1 Extracting and aligning the nonlinear latent embeddings across sessions

First we focus on learning the low-dimensional latent embeddings $\hat{u}_t^{(s)} \in \mathbb{R}^D$ of the high-dimensional neural activity $x_t^{(s)}$ that are aligned across all $M$ sessions and can be approximated with a shared LDS prior. As depicted in Fig. 1**A**, we employ an autoencoder architecture, where each session has its own neural activity encoder $f_{\theta^{(s)}} : \mathbb{R}^{N^{(s)}} \to \mathbb{R}^D$ (a deep neural network parameterized by weights $\theta^{(s)}$ for each session) that maps $x_t^{(s)}$ to $\hat{u}_t^{(s)}$.

To ensure that $\hat{u}_t^{(s)}$ encodes the shared, behaviorally relevant signals in a common space across sessions, we employ a behavior-anchored contrastive loss that draws together latent embeddings from time points with similar behaviors and separates those with dissimilar behaviors (Fig. 1**B**). Specifically, we employ the Rank-N-Contrast loss [53], which ranks every pair of embeddings $\hat{u}_t^{(s)}$ in each minibatch (total of $N$ pairs) by their behavioral distance and enforces that distances in the latent space follow the same ordering. Imposing this constraint across all pairs yields an embedding space in which inter-embedding distances faithfully mirror the corresponding continuous behavior. For notational simplicity, we drop the session superscript on $\hat{u}_t$ for the remainder of Section 3.1, since all sessions' embeddings are jointly aligned.

Given an anchor $\hat{u}_h$ with the associated behavior $y_h$, for any other embedding $\hat{u}_j$, we define a set of embeddings that have higher *ranks* than $\hat{u}_j$ in terms of the behavioral label distance with respect to the anchor as $\mathcal{S}_{h,j} := \{\hat{u}_k \mid k \neq h, d(y_h, y_k) \geq d(y_h, y_j)\}$, where $d(\cdot, \cdot)$ is the $L_1$ distance between the two behaviors. In other words, $\mathcal{S}_{h,j}$ is defined as a set of embeddings whose distance to the anchor $\hat{u}_h$ is *larger* than the distance between $\hat{u}_h$ and $\hat{u}_j$, and hence are *less* similar to the anchor $\hat{u}_h$ than $\hat{u}_j$. Define the "likelihood" associate with $\hat{u}_j$ given the anchor and the higher rank set as:

$$\mathbb{P}(\hat{u}_j \mid \hat{u}_h, \mathcal{S}_{h,j}) = \frac{\exp(\text{sim}(\hat{u}_h, \hat{u}_j)/\tau)}{\sum_{\hat{u}_k \in \mathcal{S}_{h,j}} \exp(\text{sim}(\hat{u}_h, \hat{u}_k)/\tau)} \tag{1}$$

where $\text{sim}(\cdot, \cdot)$ is the similarity measure between two latent embeddings (*e.g.*, negative $L_2$ norm) and $\tau$ is the temperature parameter. Here, the likelihood will be maximized to push $u_j$ closer to the anchor $u_h$ in the embedding space than any other embeddings in the rank set $\mathcal{S}_{h,j}$.

We define a contrastive objective based on the ranking of the continuous behavior samples:

$$\mathcal{L}_{\text{contrastive}} = -\frac{1}{N(N-1)} \sum_{h=1}^{N} \sum_{j \neq h} \log \mathbb{P}(\hat{u}_j \mid \hat{u}_h, \mathcal{S}_{h,j}) \tag{2}$$

This contrastive loss not only promotes behavioral relevance in the latent embeddings but also serves as an alignment mechanism across sessions by using behavior as a shared anchor.

In some experiments, we additionally incorporate a behavior supervision loss that explicitly encourages learned embeddings to capture the behavioral outputs:

$$\mathcal{L}_{\text{behavior}} = \frac{1}{M} \sum_{s=1}^{M} \frac{1}{L^{(s)}} \sum_{i=1}^{L^{(s)}} \text{MSE}\left(g_\gamma\left(\hat{u}_t^{(s)}\right), y_t^{(s)}\right) \tag{3}$$

where $g_\gamma(\cdot)$ is a session-invariant linear mapping from latent embeddings $\hat{u}_t$ to behavioral variables $y_t$. We evaluate the effect and discuss the limitations of this behavior supervision loss in Section 4.2.3.

## 3.2 Learning the preserved dynamical systems across sessions on the latent embedding space

To capture the preserved dynamics in the embedding space and to enable flexible casual inference, we treat the aligned latent embedding $\hat{u}_t^{(s)}$ as noisy observations of a single LDS that is shared across $M$ sessions (Fig. 1**C**). Concretely, we assume a latent state $z_t \in \mathbb{R}^{n_z}$ that evolves and generates embeddings via

$$z_t^{(s)} = A z_{t-1}^{(s)} + w_t, \quad w_t \sim \mathcal{N}(0, Q) \tag{4}$$

$$\hat{u}_t^{(s)} = C z_t^{(s)} + v_t, \quad v_t \sim \mathcal{N}(0, R) \tag{5}$$

with $A \in \mathbb{R}^{n_z \times n_z}$ (dynamics matrix), $C \in \mathbb{R}^{D \times n_z}$ (emission matrix), and the noise covariance $Q$ and $R$ shared across all sessions. We set $n_z = D$ so that the latent dynamics have the same dimensionality as the embedding space. This two-layer design enables real-time causal filtering, non-causal smoothing, and principled handling of missing data via Kalman filter, while the neural embedding captures complex structure without sacrificing tractable inference.

To causally and flexibly infer the latent state, we run a standard Kalman filter forward on the embeddings of each session. At each time $t$, the filter update gives us the causally estimated posterior state $z_{t|t}^{(s)} = \mathbb{E}\left[z_t^{(s)} \mid \hat{u}_{1:t}^{(s)}\right]$, from which we compute the filtered embedding $u_t^{(s)} = C z_{t|t}^{(s)}$.

We learn the parameters of the shared LDS, $\{A, C, Q, R\}$, by minimizing a $k$-step-ahead neural prediction loss, one of the standard methods for training LDS [22, 64–66]. The high-dimensional neural activity is reconstructed from a neural decoder $f_{\phi^{(s)}} : \mathbb{R}^D \to \mathbb{R}^{N^{(s)}}$, a deep neural network with parameters $\phi^{(s)}$ for each session:

$$\hat{x}_t^{(s)} = f_{\phi^{(s)}}\left(u_t^{(s)}\right) \tag{6}$$

where $u_t^{(s)}$ is the filtered embedding. As a result, the overall loss for the LDS is:

$$\mathcal{L}_{k\text{-step}} = \frac{1}{M} \sum_{s=1}^{M} \sum_{k=1}^{K} \sum_{t=1}^{T^{(s)}-k} \text{MSE}(\hat{x}_{t+k|t}^{(s)}, x_{t+k}^{(s)}) \tag{7}$$

### 3.3 Combining the latent embeddings and the preserved latent dynamics

Finally, we train all components of CANDY, the encoder parameters $\{\theta^{(s)}\}_{s=1}^{M}$, the neural activity decoder parameters $\{\phi^{(s)}\}_{s=1}^{M}$, the shared LDS parameters $\{A, C, Q, R\}$, and the optional behavior decoder parameters $\gamma$, in an end-to-end manner by minimizing the combined loss:

$$\mathcal{L}_{\text{CANDY}} = \mathcal{L}_{k\text{-step}} + \lambda_{\text{contrastive}}\mathcal{L}_{\text{contrastive}} + \underbrace{\lambda_{\text{behavior}}\mathcal{L}_{\text{behavior}}}_{\text{optional}} \tag{8}$$

where $\lambda_{\text{contrastive}}$ and $\lambda_{\text{behavior}}$ are the loss weights that control the relative contributions of the alignment and behavior-supervision terms, respectively. Notably, the behavior supervision loss is optional for alignment, whose effect and limitation will be discussed in detail in Section 4.2.3.

## 4 Results

We first validated CANDY using a synthetic dataset. For this initial test, a single latent dynamical system was projected onto three distinct neural observation spaces using different nonlinear mappings. This experiment demonstrated that CANDY can reliably recover the shared underlying dynamics, even when the observed data come from heterogeneous sources. Next, we evaluated CANDY's performance on two real-world neural datasets spanning species and recording modalities: two-photon calcium imaging recordings from mouse dorsolateral striatum during a wheel-turning task that we collected and made available with this paper, and publicly available electrophysiological datasets from macaque motor cortex during a center-out reaching task [1, 3, 4].

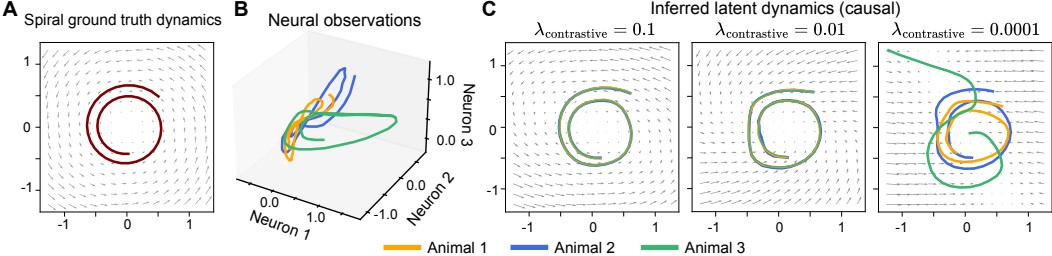

Figure 2: **Recovering shared latent dynamics from multiple heterogeneous observations. A.** Ground-truth spiral dynamics trajectory. **B.** Simulated neural observations for three animals, each generated by applying a distinct nonlinear mapping from the same underlying low-dimensional trajectory in (**A**). **C.** Latent dynamics inferred by CANDY for each animal using three different contrastive loss scales $\lambda_{\text{contrastive}}$.

### 4.1 Simulation experiment

To validate CANDY's ability to recover a common dynamical system from heterogeneous observations, we generated three synthetic "animals" as follows. Given the plausibility of spiral-like

dynamics in neural activity [67], we first simulated a two-dimensional LDS whose trajectories trace out a spiral (Fig. 2**A**) following equations:

$$z_t = A z_{t-1}, \quad A = r \cdot \begin{bmatrix} \cos(\theta) & -\sin(\theta) \\ \sin(\theta) & \cos(\theta) \end{bmatrix} \tag{9}$$

with radial decay $r = 0.99$ and rotation angle $\theta = \pi/15$. The continuous behavioral signals $y_t \in \mathbb{R}^2$ were then obtained via a shared linear mapping $y_t = W z_t$ (see Appendix A.1 for details). Finally, for each animal $s$, we applied a subject-specific nonlinear transformation $f_{\phi^{(s)}} : \mathbb{R}^2 \to \mathbb{R}^3$ with Gaussian noise to produce neural observations (Fig. 2**B**; see Appendix A.1 for details).

CANDY was trained jointly on all three datasets with latent dimensionality $D$ set to 2. Each subject-specific MLP encoder $f_\theta^{(s)} : \mathbb{R}^3 \to \mathbb{R}^2$ learns to project the noisy 3D observations back onto the shared 2D manifold, leveraging a contrastive objective between latent embeddings and behaviors. As shown in the left panel of Fig. 2**C**, the inferred trajectories of each subject, as well as the shared LDS, were correctly identified with contrastive learning despite distinct nonlinear distortions and noise. If we set the contrastive learning loss scaler smaller, the inferred trajectories exhibit weaker alignment, and the learned LDS becomes less accurate (Fig. 2**C**, middle). If we set the contrastive scaler to a much smaller value, the alignment fails and the shared LDS is not accurately identified (Fig. 2**C**, right). This suggests that behavior-anchored contrastive learning is essential to align latent embeddings across sessions and to uncover the preserved LDS.

To evaluate the generalizability of the learned LDS to a held-out animal, we generated a new neural observation from the same spiral dynamical system with a distinct nonlinear mapping (Fig. S1**A**-**B**; see Appendix A.1 for details). The LDS was then frozen, and only the encoder for the new session was trained. The resulting inferred latent dynamics for this new animal were found to be consistent with the dynamic flow field of the pretrained LDS (Fig. S1**C**), demonstrating that the LDS can generalize to new subjects.

## 4.2 Mouse wheel-turning task

We evaluated CANDY on a mouse wheel-turning dataset collected by two-photon calcium imaging (Fig. 3**A**). We recorded neural activity from the dorsolateral striatum of three adult mice (IDs: 24, 25, 26): with 4 sessions each from mice 24 and 26 and 5 sessions from mouse 25, for a total of 13 sessions and approximately 3,000 neurons ($\sim$220 neurons per session). Each session comprised roughly 200 trials, during which animals rotated a manipulandum to center a randomly initialized visual cue on an iPad screen. Calcium signals were sampled at 30Hz, resulting in 20–40 time bins per trial aligned to cue onset and one of three trial-end events: cue centered, cursor exiting the screen edge (left or right), or timeout. To assess both within- and cross-subject generalization, we split data from two mice (24 and 26) to training (60%), validation (10%), and testing (30%) sets, and held out the third mouse (25) entirely for generalization evaluation. Detailed imaging parameters and preprocessing procedures are provided in Appendix A.4.

### 4.2.1 The alignment of latent embeddings preserved behavioral structures across sessions

To assess CANDY's ability to align latent embeddings across sessions while preserving behaviorally relevant information, we compared it to two baselines: multi-session CEBRA [16], which also trains a shared embedding on all sessions jointly, and DFINE-supervised [22], which trains separate embeddings per session. We used grid search to optimize hyperparameters for both the DFINE-supervised model and multi-session CEBRA model (see Appendix B for details). We first visualized each model's latent embeddings in one 2D space via principal component analysis (PCA) (Fig. 3**B**). CANDY successfully aligns latent embeddings from multiple sessions into a single, coherent manifold, where there is no clear distinction in the latent embeddings of different sessions (Fig. 3**B**, left). Multi-session CEBRA is also able to bring latent embeddings into a common subspace with no session-specific clusters (Fig. 3**B**, middle). In contrast, DFINE-supervised produces latent embeddings that occupy distinct and disjoint regions of the latent space, indicating an absence of correspondence across sessions (Fig. 3**B**, right).

We then quantified the embedding geometry by comparing pairwise Euclidean distances in the latent embedding space to behavioral dissimilarities computed from wheel-turn velocities (Fig. 3**C**). Sorting time bins by ascending velocity magnitude, we plotted both the behavioral distance

matrix and the latent embedding distance matrix of each model to facilitate direct comparison between latent-space geometry and the true behavioral structure. The latent embeddings' distance matrix of CANDY (Fig. 3C, top right) exhibits a clear block-diagonal pattern that closely mirrors the behavioral matrix (Fig. 3C, top left) — with blocks corresponding to rightward, leftward, or stationary movements — demonstrating that CANDY's latent embeddings faithfully encode behavioral structures. By comparison, multi-session CEBRA's distance matrix (Fig. 3C, bottom left) shows only week correspondence to the behavioral structure, and DFINE-supervised (Fig. 3C, bottom right) fails to recover any block structure. To quantitatively assess how well the latent embedding geometry can reflect the task-relevant behavioral structure, we defined an alignment score as the Pearson correlation between the Euclidean distance matrices of behavior and the corresponding latent embeddings. We showed that CANDY achieves a significantly higher alignment score than multi-session CEBRA among all latent embedding dimensionalities (Fig. S2).

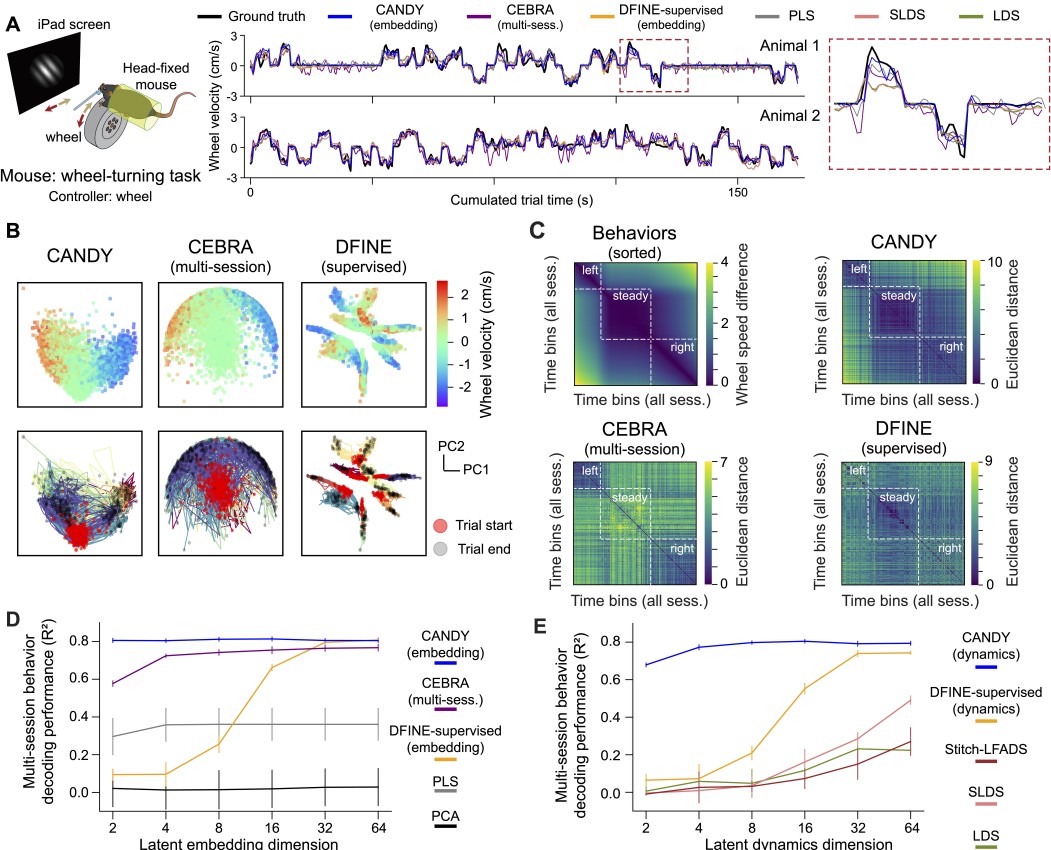

Figure 3: **CANDY aligns latent embeddings and dynamics across sessions resulting in higher behavior decoding performances. A.** Left: illustration of the mouse wheel-turning task, where positive velocity indicates rightward turning and negative velocity indicates leftward turning. Right: sample behavior traces (including a zoomed-in view) from two randomly selected testing sessions. The traces of DFINE-supervised, LDS, SLDS, and PLS are from behavior decoders trained on individual sessions separately. **B.** Top: Scatter plot of the first two principal components (PC1 vs. PC2) of the aggregated latent embeddings across all sessions, with points colored by instantaneous wheel velocity. Bottom: Same PC1–PC2 projection as the top row but colored by session IDs. Columns: (i) CANDY, (ii) multi-session CEBRA, and (iii) DFINE-supervised. **C.** Pairwise heatmaps over time bins sorted by behavioral magnitude, comparing behavioral differences (top left) and latent embedding distances of the three methods illustrated in (**B**). The latent embedding distance is defined as the Euclidean distance between the latent embeddings from two time bins with rows and columns ordered identically to the top left panel. **D-E.** Multi-session behavior decoding performance ($R^2$) evaluated with a session-agnostic behavior decoder (Appendix C.2.1) on (**D**) neural latent embeddings and (**E**) latent dynamics. Solid line: average over 8 sessions. Performance for each session is averaged over 5 seeds. Error bar: s.e.m over the 8 sessions.

These findings indicate that CANDY successfully aligns representations across sessions while capturing the inherent behavioral structure within its learned embeddings, all without the usage of a behavioral supervision loss.

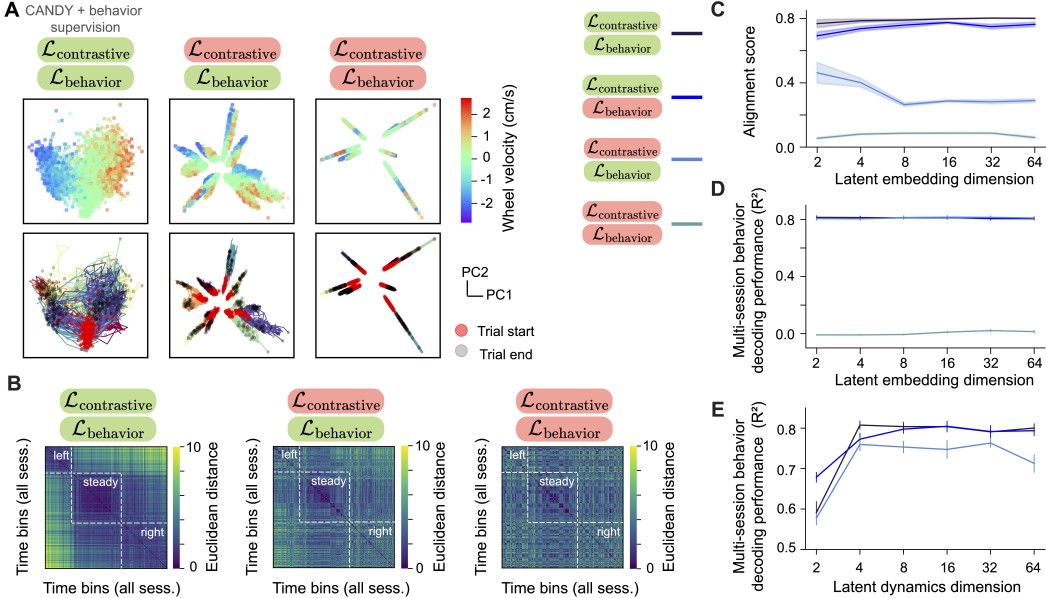

Figure 4: **Ablation study of behavior supervision and contrastive loss. A.** PC1–PC2 projections of latent embeddings aggregated across all sessions. Top row is colored by instantaneous wheel velocity; bottom row by session ID. Columns show (i) CANDY+behavior supervision loss, (ii) behavior supervision loss only (no contrastive loss), and (iii) unsupervised (neither behavior supervision loss nor contrastive loss). **B.** The pairwise distance matrix of the latent embeddings (similar as Fig. 3**C**) extracted by the three methods mentioned in panel (**A**). **C.** Alignment score for different latent embedding dimensionalities. Shaded area: s.e.m over 5 random seeds. **D-E.** Multi-session behavior decoding $R^2$ value evaluated with a session-agnostic behavior decoder (Appendix C.2.1) on (**D**) latent embeddings and (**E**) causally inferred latent dynamics. Solid lines: average over 8 sessions. Performance for each session is averaged over 5 seeds. Error bar: s.e.m. over the 8 sessions. Red background: disable the loss listed above; green background: enable the loss listed above.

### 4.2.2 Behavior decoding from latent embeddings and dynamical factors

To assess how well the aligned representations support behavior decoding, we trained a session-agnostic decoder to predict the behavioral variable at each time step from either the latent embeddings or the inferred latent dynamics across all sessions (Appendix C.2.1).

CANDY demonstrates high behavior decoding performance on latent embeddings across all sessions with much lower dimensions needed than other methods (Fig. 3**D**). These results show that CANDY can uncover a behaviorally relevant low-dimensional structure, supporting the hypothesis that a shared latent embedding space exists across sessions. Across all latent dimensions, CANDY yields higher decoding accuracy than multi-session CEBRA [16], DFINE-supervised [22], partial least squares regression (PLS), and PCA, with markedly larger gains at moderate dimensions and a diminishing margin at very large dimensions where DFINE-supervised is essentially on par.

Next, we evaluated the decoding performance on the causally inferred latent dynamics extracted by CANDY, $\{z_{1:T^{(s)}}\}_{s=1}^{M}$, to test the effectiveness of the preserved LDS. Similarly, the behavior decoding performance plateaus at only 8 latent dimensions (Fig. 3**E**). In contrast, latent dynamics of models built on unaligned embeddings, including DFINE-supervised [22], Stitch-LFADS [14], LDS, and SLDS [26], all underperform CANDY across all dimensionalities and require substantially more dimensions to converge. We verified that all models correctly learned neural dynamics within their respective sessions by evaluating the decoding performance with session-specific decoders (Fig. S3; see Appendix C.2.2 and D.3 for details). Furthermore, we demonstrated that the extracted LDS from

CANDY can forecast future behaviors with the pretrained behavior decoder, substantially better than DFINE-supervised (Fig. S4; see Appendix D.4 for details).

### 4.2.3 Behavior supervision loss cannot replace contrastive loss for alignment

To assess the impact of behavior supervision on embedding and dynamics alignment, we compared four training regimes: (i) CANDY ($\lambda_{\text{behavior}} = 0$), (ii) CANDY with behavior-supervision loss ($\lambda_{\text{behavior}} = 1$), (iii) CANDY without contrastive learning ($\lambda_{\text{contrastive}} = 0$) and (iv) a fully ablated model ($\lambda_{\text{contrastive}} = \lambda_{\text{behavior}} = 0$).

We showed that only with contrastive learning can the latent embeddings be aligned and capture the continuous behavioral structure (Fig. 3**B**-**C** and Fig. 4**A**-**B**). Contrastive learning enables the models to achieve the highest alignment score and augmenting CANDY with behavior supervision can further increase the alignment score (Fig. 4**C**), and thus better reflect the behavioral structure in the latent embedding space.

However, behavior supervision loss alone cannot replace contrastive learning for alignment. The model with only behavior supervision loss fails to align latent embeddings across sessions (Fig. 4**A**, middle), breaks the block-diagonal behavioral structure (Fig. 4**B**, middle), and achieves a substantially lower alignment score (Fig. 4**C**). Interestingly, all three variants, the original contrastive-only model, the combined contrastive + behavior model, and the behavior-only model, achieve similar behavior decoding performance from the latent embeddings (Fig. 4**D**). Nevertheless, when we decode behavior from the inferred latent dynamics, the behavior-only model performs poorly (Fig. 4**E**), indicating that correct latent dynamics cannot be learned without embedding alignment via contrastive learning.

Finally, in an ablation experiment where we removed both losses entirely, the latent embeddings collapse into misaligned and behaviorally uninformative spaces (Fig. 4**C**-**D**). Together, these results underscore that while behavior supervision sharpens alignment, it cannot replace the contrastive loss: aligning embeddings via contrastive learning is critical not only for embedding-level decoding but also for uncovering the preserved LDS across sessions.

In conclusion, the contrastive loss is indispensable for initial, cross-session embedding alignment, while behavior supervision loss further enhances that alignment once established.

### 4.2.4 Generalization of the preserved LDS to held-out sessions.

To assess CANDY's ability to generalize to a novel subject, we first pretrained the model (with latent dimension $D = 8$) on 8 sessions from two animals. We froze the shared LDS and the behavior decoder from the pretrained model, and trained only the subject-specific encoder and neural decoder on a held-out third animal (5 sessions).

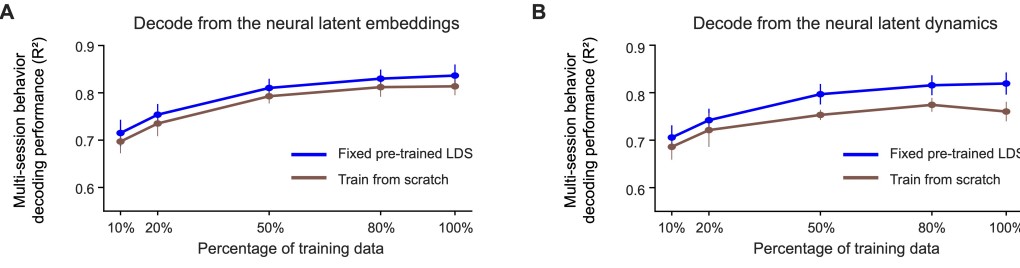

Figure 5: **Generalization of CANDY to held-out sessions. A.** Decoding performance (measured by $R^2$) using the latent embeddings $\hat{u}_t$ as a function of the percentage of held-out session training data. Blue curve: results with the fixed pretrained LDS and session-agnostic behavior decoder (Appendix C.2.1) from CANDY; brown curve: training the model from scratch on the held-out data. **B.** Same comparison for decoding from the inferred latent dynamics $z$. Solid line: average over 5 sessions and each session's performance is averaged over 5 model seeds. Error bar: s.e.m. over the 5 sessions.

With the pretrained LDS and behavior decoder, training with just 10% of the held-out data already yields $R^2 \approx 0.7$. Increasing the fraction to 50% pushes performance to $R^2 \approx 0.8$. Crucially, at every fraction—including 100%—the model with pretrained LDS and behavior decoder outperforms a

model trained from scratch on the same sessions. Similarly, behavior decoding from latent dynamics (Fig. 5**B**) achieves relatively high $R^2 \approx 0.8$ with 50% of the data. The fact that our frozen LDS generalizes seamlessly, producing high decoding accuracy without re-estimating the LDS parameters, confirms that the temporal structure learned by CANDY is truly preserved across subjects.

### 4.3 Monkey center-out reaching task

We also validated CANDY on a dataset of a different species, a different recording modality and a different behavioral task (see Appendix A.5 for dataset and preprocessing details). Spikes trains in the motor cortex of two monkeys are recorded during a center-out reaching task (Fig. S5**A**). CANDY achieves higher decoding performance of the hand velocity in $x$ and $y$ directions on latent embeddings over 40 sessions than multi-session CEBRA, DFINE-supervised, PLS and PCA (Fig. S5**A**-**B**) and higher decoding performance on latent dynamics than DFINE-supervised, LDS and SLDS (Fig. S5**C**) using a session-agnostic behavior decoder. Moreover, we demonstrated that the extracted LDS from CANDY can forecast future behaviors with the pretrained behavior decoder, substantially better than DFINE-supervised (Fig. S6; Appendix D.4). Additionally, CANDY captures the underlying behavioral structure in the latent embedding space suggested by a substantially higher alignment score than multi-session CEBRA (Fig. S7). Furthermore, the pretrained LDS generalizes to two held-out sessions, achieving similar decoding performance on both latent embeddings and latent dynamics as the model trained from scratch on all held-out sessions (Fig. S8). These confirm that CANDY is able to align the latent embeddings and extract the preserved dynamics from various species and recording modalities.

## 5 Discussion

In this work, we introduced CANDY, an end-to-end framework that integrates rank-based contrastive alignment with dynamical modeling to uncover a preserved LDS shared across sessions and subjects. The alignment of the latent embeddings via rank-based contrastive learning preserves the intrinsic continuity in the task-relevant behavioral data. This enables the temporal structure within the embedding space to be faithfully modeled by an interpretable dynamical system. Across a synthetic dataset with distinct nonlinear observations and two real-world datasets, two-photon calcium imaging from mouse striatum, collected by ourselves, and electrophysiological spike trains from macaque motor cortex, CANDY consistently: (i) aligned latent embeddings across sessions, (ii) preserved the intrinsic structure of the continuous behavior in the latent embeddings, (iii) achieved accurate behavior decoding on both aligned latent embeddings and shared latent dynamics, and (iv) learned a preserved LDS that is generalizable to unseen sessions.

Looking ahead, several directions could further enhance CANDY's versatility and impact. First, incorporating nonlinear dynamics could capture richer temporal patterns beyond the linear Gaussian assumption. While CANDY's expressive encoders can project complex dynamics onto a linearly tractable latent space (Fig. S9; see Appendix A.2 for details), this risks obscuring the true underlying nonlinear mechanisms. Second, while extracting preserved neural computations is a primary goal, identifying the complementary subject-distinctive dynamical strategies [68] is useful for understanding the underlying strategy that govern motor behaviors. Third, while CANDY demonstrates strong generalization of the pretrained LDS and behavior decoder to unseen sessions, generalization to completely novel behavioral states poses a fundamental challenge for both CANDY and existing approaches. Specifically, we held out one moving direction of the monkey center-out reaching task entirely for testing, and demonstrated that both CANDY (test $R^2$: mean -0.1209, s.e.m. 0.0319) and CEBRA (test $R^2$: mean -0.2138, s.e.m. 0.0248) failed to generalize to the unseen behavior category, which requires future exploration. Fourth, though CANDY primarily focused on linear behavior mapping, extending this to nonlinear behavior mapping or even subject-specific behavior readout is an interesting future direction. Finally, beyond extracting shared dynamics, CANDY shows promise as a diagnostic tool for discovering dynamical heterogeneity. Our synthetic experiments (Fig. S10; see Appendix A.3 for details) indicate that when presented with sessions from disparate underlying dynamics, the model provides clear signals of alignment failure, such as unaligned latent trajectories and a marked performance drop in the session-agnostic decoder. The crucial next step is to validate this capability on experimental data where heterogeneity is suspected.

## Acknowledgments and Disclosure of Funding

Dr. Mark Schnitzer gratefully acknowledges funding from the Simons Foundation Collaboration on the Global Brain and the Vannevar Bush Faculty Fellowship Program of the U.S. Department of Defense. Yiqi Jiang and Fatih Dinc are supported by Stanford University's Mind, Brain, Computation and Technology program under the Stanford Wu Tsai Neuroscience Institute. Kaiwen Sheng is supported by a Stanford Bioengineering department fellowship. Fatih Dinc is supported in part by the National Science Foundation under Grants PHY-2309135 and 2313150, and the Gordon and Betty Moore Foundation Grant No. 2919.02 to the Kavli Institute for Theoretical Physics (KITP). We thank Lili Tai and Yu Jiang for computing resources support and Dr. Itamar Landau for fruitful discussions.

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

## Appendix

The appendix of this paper is structured as follows:

- Section A introduces the evaluated datasets.
- Section B details the compared baselines in our experiments.
- Section C summarizes the training configuration we applied in this paper.
- Section D presents more comprehensive evaluation results.
- Section E discloses compute resources used for this paper.

## A  Dataset

### A.1  Simulated spiral dynamics

We generated latent trajectories by simulating the spiral dynamics from Eq. 9 for 50 time steps starting from an initial state $z_0 = (0.5, 0.5)$. For each trial, we added independent Gaussian noise $\epsilon_z \sim \mathcal{N}(0, \sigma_z^2 I_z)$ with $\sigma_z = 0.01$ to the resulting latent states.

The behavior variable $y_t$ was generated from the latent state $z_t$ via linear mapping:

$$y_t = W \cdot z_t + \epsilon_y \tag{S1}$$

where $W$ is defined as an identity matrix of size $\mathbb{R}^{2 \times 2}$, and $\epsilon_y$ is a gaussian noise sampled from $\mathcal{N}(0, \sigma_y I_2)$ with $\sigma_y = 0.001$.

The neural observation $x_t$ for each session $s$ were generated via:

$$x_t^{(s)} = f_{\phi^{(s)}}(z_t) + \epsilon_x^{(s)} \tag{S2}$$

with the nonlinear mapping for each session defined as

$$f_{\phi^{(1)}} = \begin{bmatrix} z_t[0] \\ z_t[1] \\ \sin(2 \cdot z_t[0]) + \cos(2 \cdot z_t[1]) \end{bmatrix} \tag{S3}$$

$$f_{\phi^{(2)}} = \begin{bmatrix} z_t[0] \\ 0.87 z_t[0] - 0.5 \sin(2 \cdot z_t[0]) - 0.5 \cos(2 \cdot z_t[1]) \\ 0.5 z_t[0] + 0.87 \sin(2 \cdot z_t[0]) + 0.87 \cos(2 \cdot z_t[1]) \end{bmatrix} \tag{S4}$$

$$f_{\phi^{(3)}} = \begin{bmatrix} 0.71 z_t[0] + 0.71 \sin(2 \cdot z_t[0]) + 0.71 \cos(2 \cdot z_t[1]) \\ z_t[1] \\ -0.71 z_t[0] + 0.71 \sin(2 \cdot z_t[0]) + 0.71 \cos(2 \cdot z_t[1]) \end{bmatrix} \tag{S5}$$

The observation noise $\epsilon_x^{(s)}$ was sampled independently for each session from $\mathcal{N}(0, \sigma_x I_3)$ with $\sigma_x = 0.1$.

For the held-out animal in Fig. S1, the nonlinear mapping is defined as

$$f_{\phi^{(4)}} = \begin{bmatrix} z_t[0] \\ z_t[1] \\ \sin(2 \cdot z_t[0]) + \sin(2 \cdot z_t[1]) \end{bmatrix} \tag{S6}$$

### A.2  Simulated limit cycle

We simulated the limit cycle dynamics using the Van der Pol oscillator as

$$\begin{aligned} \frac{dz_1}{dt} &= z_2 \\ \frac{dz_2}{dt} &= \mu(1 - z_1^2) z_2 - z_1 \end{aligned} \tag{S7}$$

where $\mu = 1.0$. We numerically integrated these equations from an initial condition of $z_0 = (0.05, 0.05)$ for 200 time steps to generate the latent limit cycle trajectory. Independent Gaussian noise $\epsilon_z \sim \mathcal{N}(0, \sigma_z^2 I_z)$ with $\sigma_z = 0.01$ was added to the resulting latent trajectories for each trial.

The behavioral variable $y_t$ was generated according to Eq. S1 with $\sigma_y = 0.001$. The neural observation for each session was generated according to Eq. S2 with the session-specific nonlinear mappings $f_{\phi^{(1)}}, f_{\phi^{(2)}}$, and $f_{\phi^{(3)}}$ corresponding to those defined in Eqs. S3-S5, respectively. The observation noise $\epsilon_x^{(s)}$ was sampled independently for each session from $\mathcal{N}(0, \sigma_x I_3)$ with $\sigma_x = 0.1$.

### A.3 Simulated multiple dynamics

We simulated three sessions from two distinct dynamical systems: spiral dynamics as described in Appendix A.1, and limit cycle dynamics as described in Appendix A.2. The neural observations of two sessions were generated from the spiral dynamics $z_t^{\text{spiral}}$ using the nonlinear mappings of Eq. S3 and S4. The neural observation of the third session was generated from the limit cycle dynamics $z_t^{\text{cycle}}$ using a distinct nonlinear neural observation mapping:

$$f_{\phi^{(3)}} = \begin{bmatrix} z_t[0] \\ z_t[1] \\ \cos(2 \cdot z_t[0]) + \sin(2 \cdot z_t[1]) \end{bmatrix} \tag{S8}$$

For all three sessions, the observation noise $\epsilon_x^{(s)}$ was sampled independently from $\mathcal{N}(0, \sigma_x I_3)$ with $\sigma_x = 0.1$ and the behavioral variable $y_t$ was generated according to Eq. S1 with $\sigma_y = 0.001$.

### A.4 Calcium imaging data acquisition and preprocessing

Calcium imaging was performed using a custom-built two-photon microscope, with excitation at 920 nm and fluorescence emission peaking at 512 nm (jGCaMP8m). The imaging frame size was $512 \times 512$ pixels with a spatial resolution of 0.8 $\mu$m per pixel, and data were sampled at 30 Hz. A GRIN lens (4 mm length, 1 mm diameter; Inscopix) was chronically implanted over the right hemisphere to target the right dorsolateral striatum. To express jGCaMP8m in medium spiny neurons, an AAV2/PHP.eB-CaMKII-jGCaMP8m virus ($5.2 \times 10^{12}$ gc/mL) was injected at stereotaxic coordinates of AP +1.0 mm, ML + 2.25 mm, and DV -2.4 mm from the brain surface. Following data acquisition, the imaging movies underwent preprocessing, including fluorescence decay correction, motion correction, and z-scoring. Neuronal regions of interest (ROIs) were automatically extracted using the EXTRACT algorithm [69] implemented in MATLAB 2022a, followed by manual curation to ensure accurate cell identification.

### A.5 Monkey reaching task data preprocessing

We used electrophysiological recordings from the motor cortex collected during a center-out reaching task [1, 3, 4], in which monkeys were trained to move a manipulandum toward one of eight targets displayed on a screen. The neural and behavioral data were obtained from: https://doi.org/10.48324/dandi.000688/0.250122.1735. For training, we used 40 sessions from this task recorded from subjects C and M. 2 sessions from these two subjects were held out entirely for evaluating the generalization of the pretrained LDS. A fixed 100 ms delay between neural activity and movement was assumed across all sessions. Spiking activity was binned in 20 ms intervals, smoothed with a 50 ms causal Gaussian filter, and then downsampled by a factor of two to obtain firing rates. Neurons with firing rates below 0.1 Hz were excluded, and all neural data were aligned to the go cue onset. The ratio of training, validation and testing datasets is 6:1:3.

## B Baselines

For each model, we selected the hyperparameters using grid search. We trained each model five times with five random seeds. The performance was averaged over all sessions and all seeds.

### B.1 LDS and SLDS

We leveraged publicly available code (https://github.com/lindermanlab/ssm) and set the dynamics matrix to be Gaussian and the emission matrix to be orthogonal Gaussian. The number of

states in the SLDS model is selected between 1, 2, and 3 based on the behavior decoding performance in the validation set. We used Laplace EM algorithm with 50 iterations for training and structured mean field as the variational posterior for both LDS and SLDS.

## B.2 multi-session CEBRA

CEBRA [16] learns behaviorally relevant embeddings by applying a contrastive loss to finite windows of neural traces, thus non-causal. Because it relies on a feedforward convolutional network with a fixed receptive field, CEBRA does not capture explicit temporal dynamics beyond the input window. At each gradient step, positive and negative pairs are drawn using a mixed time–behavior conditional sampler, which ensures that positive samples share the same discrete behavioral label and are also constrained within a defined time offset. We selected the model parameters using grid search on the hyperparameters. Eventually, the model is trained via the publicly available package (https://cebra.ai/docs/installation.html). We trained offset10-model for 5000 iterations, with a batch size for contrastive learning of 2048, cosine distance as the contrasive distance metric, using the Adam optimizer of learning rate $3 \times 10^{-4}$ on both mouse and monkey datasets.

## B.3 DFINE-supervised

DFINE-supervised [22] learns behaviorally-relevant nonlinear embeddings and infers an LDS for a single session. The nonlinear embeddings are learned via an MLP. We used the publicly available code (https://github.com/ShanechiLab/torchDFINE). We tuned the hyperparameters of DFINE-supervised on all datasets via grid search. For the mouse wheel-turning tasks, the model is trained with 4-steps-ahead prediction loss, Adam optimizer [70] with learning rate of $2 \times 10^{-3}$, $\ell_2$ regularization scale of $5 \times 10^{-3}$, behavior loss scale $\lambda_{\text{behavior}} = 1$, encoder and decoder MLP with hidden layers size [64, 16], leaky ReLU activation function and batch size of 32 for 500 epochs. For the monkey center-out reaching task, the model is trained with 2-steps-ahead prediction loss, Adam optimizer with learning rate of $2 \times 10^{-3}$, $\ell_2$ regularization scale of $2 \times 10^{-2}$, behavior loss scale $\lambda_{\text{behavior}} = 1$, encoder and decoder MLP with hidden layers size [32, 32], mish activation function and batch size of 32 for 200 epochs.

## B.4 Stitch-LFADS

Stitch-LFADS [14] inferred a common RNN-based dynamics across all sessions by leveraging session-specific linear read-in and linear read-out for the neural recordings. We used the publicly available code (https://github.com/arsedler9/lfads-torch). We tuned the hyperparameters via grid search. For the mouse wheel-turning task, the model is trained with Adam optimizer [70] with learning rate $4 \times 10^{-3}$, learning rate decay as 0.998, number of epochs as 3000, gradient clip as 5, dropout rate 0.01, dimensionality of the encoding data as 64, time steps to be used for only the initial condition encoder as 0, dimensionality of the initial condition encoder as 4, dimensionality of the controller input encoder as 64, lag of the controller input as 1, dimensionality of the controller as 16, dimensionality of the controller output as 16, dimensionality of the initial condition as 32, dimensionality of the generator as 32, prior distribution for the controller output and for the initial condition as multivariate normal distribution, scaling factor for the KL divergence regularization of the initial condition as $10^{-3}$, scaling factor for the KL divergence regularization of the controller output as $10^{-3}$, $\ell_2$ regularization scale as $10^{-4}$.

# C Training configurations

## C.1 Model hyperparameters

We tuned our model with grid search for all datasets, including the simulation dataset, the mouse dataset, and the monkey dataset.

For the simulation spiral dynamics dataset, we used Adam optimizer [70] with learning rate of $2 \times 10^{-3}$, $\ell_2$ regularization scale of $5 \times 10^{-3}$, encoder and decoder MLP with hidden sizes of [3, 2], 500 training epochs, leaky ReLU activation function, batch size of 32, 2048 time points for contrastive learning, contrastive temperature $\tau = 0.1$, contrastive scale $\lambda_{\text{contrastive}} \in \{0.1, 0.01, 0.0001\}$, $\ell_1$ as the

contrastive distance measure, $\ell_2$ as the contrastive similarity measure, 4-steps-ahead prediction loss, and $\lambda_{\text{behavior}} = 0$, *i.e.*, without behavior supervision loss.

For the mouse wheel-turning dataset, we used Adam optimizer [70] with learning rate of $2 \times 10^{-3}$, $\ell_2$ regularization scale of $5 \times 10^{-3}$, encoder and decoder MLP with hidden sizes of [64, 16], 500 training epochs, leaky ReLU activation function, batch size of 32, 2048 time points for contrastive learning, contrastive temperature $\tau = 0.2$, , contrastive scale $\lambda_{\text{contrastive}} = 0.1$, $\ell_1$ as the contrastive distance measure, $\ell_2$ as the contrastive similarity measure, 4-steps-ahead prediction loss, and $\lambda_{\text{behavior}} = 0$ for Fig. 3 and $\lambda_{\text{behavior}} = 1$ for Fig. 4 when the behavior supervision loss is enabled.

For the monkey center-out reaching dataset, we used Adam optimizer [70] with learning rate of $2 \times 10^{-3}$, $\ell_2$ regularization scale of $2 \times 10^{-2}$, encoder and decoder MLP with hidden sizes of [32, 32], 200 training epochs, mish activation function, batch size of 32, 2048 time points for contrastive learning, contrastive temperature $\tau = 0.5$, contrastive scale $\lambda_{\text{contrastive}} = 0.1$, $\ell_1$ as the contrastive distance measure, $\ell_2$ as the contrastive similarity measure, $\lambda_{\text{behavior}} = 1$, and 2-steps-ahead prediction loss.

### C.2   Evaluating behavior decoding performance of latent embeddings and latent dynamics

We defined two types of behavior decoding performance evaluations: **session-agnostic** decoding (universal, trained across all sessions) and **session-specific** decoding (trained independently per session).

#### C.2.1   Session-agnostic behavior decoding

We concatenated latent embeddings or latent dynamics across all sessions into a design matrix $\mathbf{U} \in \mathbb{R}^{N \times D}$, where $N = \sum_{s=1}^{M} \sum_{i=1}^{L^{(s)}} T^{(s,i)}$. The corresponding behavioral targets $\mathbf{y} \in \mathbb{R}^{N \times n_y}$ were concatenated in the same order. Each column of $\mathbf{U}$ was standardized to zero mean and unit variance, and we fit an $\ell_1$-regularized linear model (Lasso):

$$\hat{\beta}_u \; = \; \arg\min_{\beta} \; \frac{1}{N} \big\| \mathbf{y} - \mathbf{U}\beta \big\|_2^2 \; + \; \lambda \|\beta\|_1,$$

where $\lambda$ was selected via 10-fold cross-validation on the training data. Decoding performance was then reported on the held-out test set.

#### C.2.2   Session-specific behavior decoding

For each session $s \in \{1, \dots M\}$, we concatenate latent embeddings or latent dynamics across all sessions into a design matrix $\mathbf{U}^{(s)} \in \mathbb{R}^{N^{(s)} \times D}$, where $N^{(s)} = \sum_{i=1}^{L^{(s)}} T^{(s,i)}$. The corresponding behavioral targets $\mathbf{y}^{(s)} \in \mathbb{R}^{N^{(s)} \times n_y}$ were concatenated in the same order. Each column of $\mathbf{U}$ was standardized to zero mean and unit variance, and we fit an $\ell_1$-regularized linear model (Lasso):

$$\hat{\beta}^{(s)} \; = \; \arg\min_{\beta} \; \frac{1}{N^{(s)}} \big\| \mathbf{y}^{(s)} - \mathbf{U}^{(s)}\beta \big\|_2^2 \; + \; \lambda \|\beta\|_1,$$

## D   Additional experimental results

### D.1   Simulation experiment on nonlinear dynamics

To show that CANDY can align the latent dynamical factors trajectories generated from a nonlinear dynamical system, we simulated three distinct noisy neuronal observations generated from the same underlying limit cycle dynamics (see Appendix A.2 for details). The latent dynamical factors are aligned and correctly inferred even though CANDY uses LDS to model the dynamics (Fig. S9), showing the ability of the model to characterize potential nonlinearities in the dynamics with the nonlinear encoder design choice.

### D.2   Simulation experiment on differentiation of different latent dynamics

To assess whether CANDY can detect when sessions employ different underlying dynamical strategies, we conducted a controlled synthetic experiment. We generated three sessions of neural activity with

identical linear behavioral readouts but distinct latent dynamics. Two sessions were simulated using a spiral attractor, while the third followed a limit-cycle attractor (see Appendix A.3 for details).

We trained CANDY jointly on all three sessions. The inferred latent trajectories revealed that the two spiral sessions were consistently aligned, whereas the limit-cycle session was not aligned with them (Fig. S10**A** Left). On the other hand, when the three sessions are all generated from the spiral dynamics, the latent trajectories are aligned (Fig. S10**A** Right). This indicates that CANDY can aligned sessions with shared dynamics while isolating sessions that deviate from the common structure.

We next compared behavioral decoding using both session-agnostic decoder and session-specific decoders (Appendix C.2). When latent dynamics were not shared across sessions (spiral vs. limit-cycle), session-specific decoders achieved significantly higher accuracy than the session-agnostic decoder ($p < 10^{-4}$, Wilcoxon signed-rank test; Fig. S10**B**). In contrast, when all three sessions were generated from spiral dynamics, the two decoding strategies achieved nearly identical performance (no significant difference, Wilcoxon signed-rank test; Fig. S10**B**).

These results demonstrate that CANDY not only recovers shared latent dynamics but may also serve as a diagnostic tool to flag sessions (or subjects) that adopt different dynamical strategies. We view this as a promising extension of our framework for large-scale, cross-subject neural analyses.

### D.3 Behavior decoding performance of single sessions

For the mouse wheel-turning task, we additionally trained session-specific behavior decoders on individual datasets (see Appendix C.2.2). This analysis serves as a control to verify that all models were trained and evaluated correctly (Fig. S3). The results further support our main evaluation strategy, where session-agnostic behavior decoding (Appendix C.2.1) provides a principled measure of alignment across sessions.

### D.4 Estimated dynamics flow field and forecasting results

To assess whether CANDY correctly learned the preserved dynamics across sessions, we performed a neural-to-behavior forecasting evaluation on both the mouse wheel-turning dataset and the monkey center-out-reaching dataset. By selecting the best model parameters and initialization seed using the validation set, we showed that the trajectory of latent dynamical factors from each trial ($z_{1:T}$) follows the dynamical flow field closely by projecting the original latent dimensions onto the top two principal components for visualization (Fig. S4**A** & Fig. S6**A**). We then performed causal k-step forecasting using the trained LDS and the behavior decoder, applied to filtered latent factors to ensure strict causality. These analyses demonstrated that the learned LDS in CANDY enables effective neural-to-behavior forecasting, consistently outperforming DFINE-supervised (Fig. S4**B** and Fig. S6**B**; Appendix C.2.1; Appendix C.2.2).

### D.5 Random shuffling behavior decoding performance

To confirm that CANDY's embeddings indeed capture genuine neural–behavioral relationships rather than model artifacts, we conducted a shuffling control experiment. Specifically, we randomly permuted the behavioral data relative to the neural recordings 20 times to disrupt their correspondence. Under these conditions, the model failed to converge: the training loss remained high, the learned embeddings lacked coherent structure, and no interpretable session alignment emerged (Fig. S11**A-B**). Furthermore, the behavior decoding $R^2$ value dropped to near zero when evaluated across sessions, both from neural latent embeddings and from latent dynamics (Fig. S11**C-D**). These findings demonstrate that CANDY's ability to extract structured, behaviorally meaningful embeddings critically depends on the true correspondence between neural activity and behavior.

### D.6 Analysis of temporal structure in the latent space

To examine whether CANDY's latent space captures meaningful temporal structure beyond behavioral alignment, we analyzed the similarity between embeddings across time within individual trials. For each trial, we computed the pairwise Euclidean distance between embeddings at all time bins, resulting in a temporal similarity heatmap (Fig. S12**A**). These heatmaps show that embeddings from

temporally adjacent bins remain close in latent space, while embeddings from bins further apart in time become progressively more distant.

To quantify this relationship, we averaged Euclidean distances as a function of temporal separation across all trials and sessions (Fig. S12**B**). The results reveal a clear monotonic increase in distance with temporal lag, indicating that CANDY produces embeddings that evolve smoothly over time. This temporal continuity is consistent with the latent dynamical system (LDS) component of the model and demonstrates that the embeddings preserve not only behavioral similarity but also intrinsic trial structure.

# E   Resources for reproducibility

Most experiments performed in this work were performed on Stanford University Sherlock CPU clusters. Running the experiment of contrastive batch size equal to 2048 requires 120GB memory. Some experiments were performed on a desktop with an Intel(R) Core(TM) i9-10900X CPU with 10 physical cores, 256G RAM, and an NVIDIA GeForce RTX 4070 Ti GPU, or a desktop with an AMD Ryzen 9 9950X 16-core 32-thred CPU, 192G RAM, and an NVIDIA GeForce RTX 4070 Ti GPU.

# Supplementary Figures

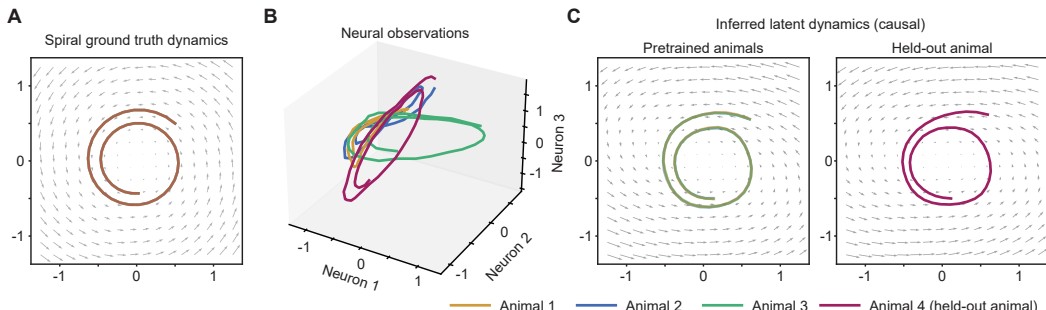

Figure S1: **The pretrained LDS generalizes to unseen synthetic animals. A.** Ground-truth spiral dynamics trajectory. **B.** Simulated neural observations for three pretrained animals and one held-out animal, each generated by applying a distinct nonlinear mapping from the same underlying low-dimensional trajectory in (**A**). **C.** The dynamic flow field of the LDS pretrained on the three animals and their corresponding latent dynamics inferred by the pretrained CANDY (left) and the latent dynamics of the held-out animal inferred by the fine-tuned CANDY (right).

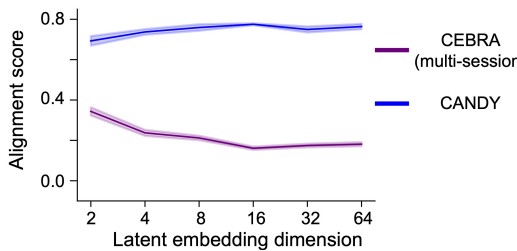

Figure S2: **Quantifying the alignment of latent embeddings to behavioral structure.** We compared the alignment score for CANDY (blue) and multi-session CEBRA (purple) using the mouse wheel-turning dataset. Solid lines: the average alignment score over 5 random seeds. Shaded areas: s.e.m. over 5 random seeds.

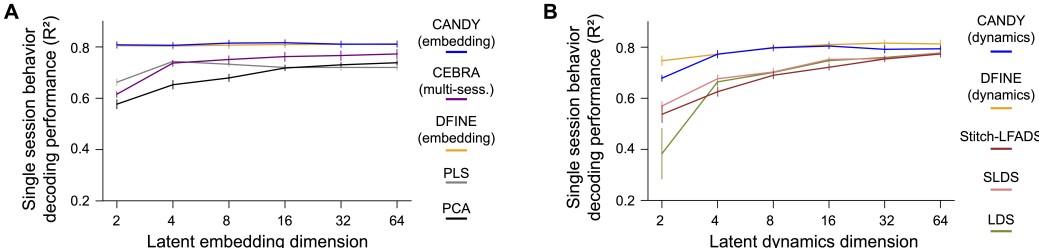

Figure S3: **Behavior decoding performance on single session. A.** The behavior decoding $R^2$ on single sessions by training a behavior decoder for each session separately. **B.** The behavior decoding $R^2$ on single sessions by training a behavior decoder for each session separately. Solid line: average over 8 sessions; for each session, model performance is averaged over 5 random seeds. Error bar: s.e.m over 8 sessions.

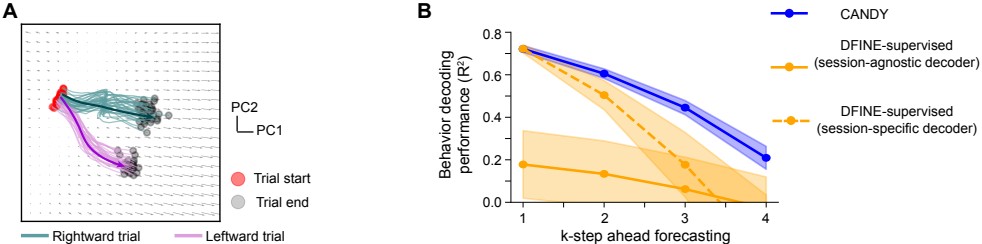

Figure S4: **CANDY successfully learned the preserved dynamics enabling high neural-to-behavior forecasting in the mouse wheel-turning dataset. A.** Visualization of the dynamics flow field in the latent subspace, projected onto the first two PCs of both the latent trajectories ($z_{1:T}$) and the learned flow field. Trial types are color-coded (green: rightward trial; pink: leftward trial), with the trial start (red dot) and the trial end (gray dot) marked. The original latent dimension is four. **B.** Solid line: neural–to-behavior forecasting using a session-agnostic decoder (Appendix C.2.1) across all sessions (solid line). Dash line: neural-to-behavior forecasting using session-specific decoders (Appendix C.2.2). Performance of CANDY (blue) is compared with DFINE-supervised (orange). Line: average over 8 sessions. Shaded area: s.e.m over 8 sessions.

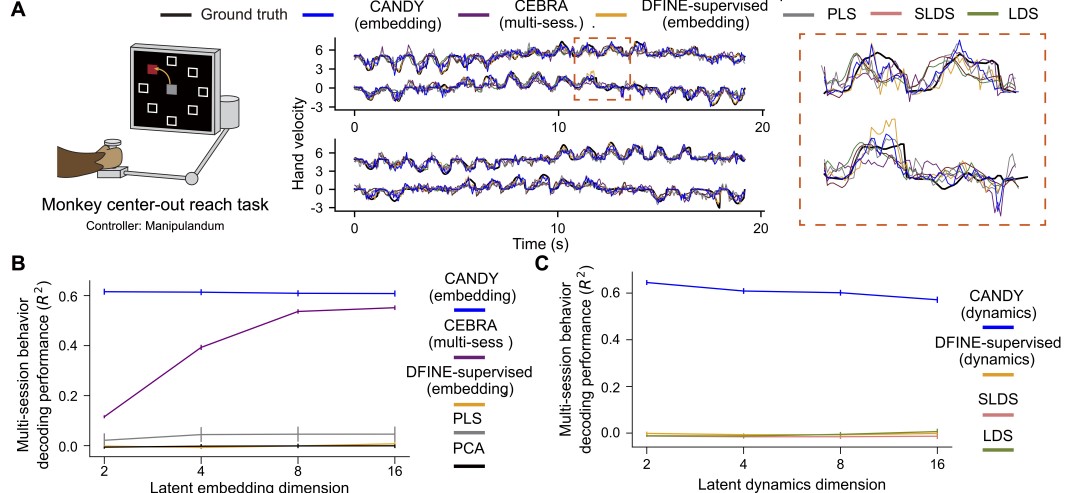

Figure S5: **CANDY achieves higher behavior decoding performances in the monkey center-out reaching task. A.** Left: illustration of the monkey center-out reaching task. Right: sample behavior traces (including a zoom-in view) on the testing datasets from two randomly selected sessions. The model and the behavior decoder of DFINE-supervised, LDS, SLDS, and PLS are trained on individual sessions separately. Top and bottom rows in each subplot are the velocity in x and y directions respectively. **B-C.** Multi-session behavior decoding performance ($R^2$) evaluated with a universal linear behavior decoder (Appendix C.2.1) on (**B**) latent embeddings and (**C.**) latent dynamics. Solid line: average over 40 sessions; for each session, model performance is averaged over 5 random seeds. Error bar: s.e.m. over the 40 sessions.

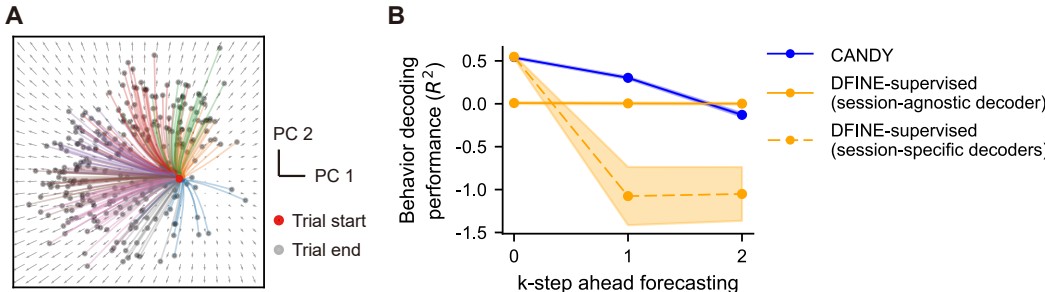

Figure S6: **CANDY successfully learned the preserved dynamics enabling high neural-to-behavior forecasting in the monkey center-out reaching dataset. A.** Visualization of the dynamics flow field in the latent subspace, projected onto the first two PCs of both the latent trajectories ($z_{1:T}$) and the learned flow field. Eight target directions of the task are color-coded, with the trial start (red dot) and the trial end (gray dot) marked. The original latent dimension is two. **B.** Solid line: neural–to-behavior forecasting using a session-agnostic decoder (Appendix C.2.1) across all sessions (solid line). Dash line: neural-to-behavior forecasting using session-specific decoders (Appendix C.2.2). Performance of CANDY (blue) is compared with DFINE-supervised (orange). Line: average over 40 sessions. Shaded area: s.e.m over 40 sessions.

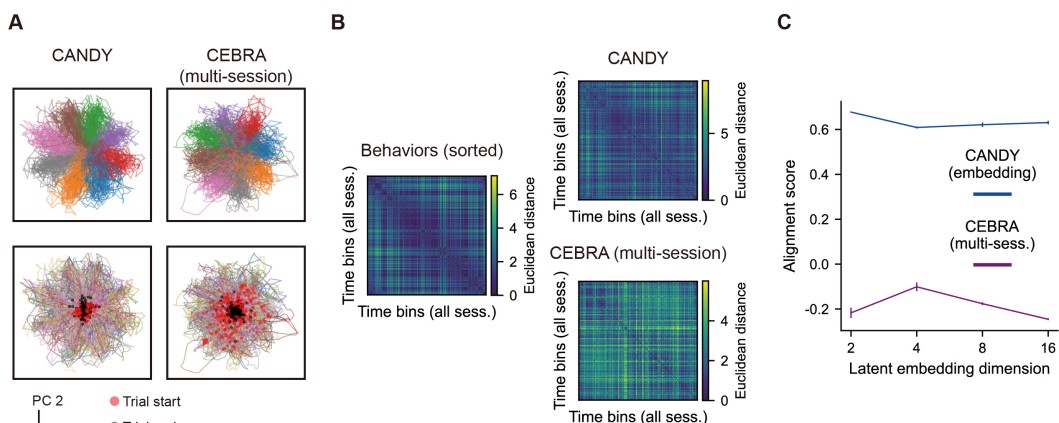

Figure S7: **CANDY aligns latent embeddings across sessions in the monkey center-out reaching task. A.** Top: Scatter plot of the first two principal components (PC1 vs. PC2) of the latent embeddings across 40 sessions, with points colored by the target direction. Bottom: Same PC1–PC2 projection as the top row but colored by session IDs. Columns: (i) CANDY, (ii) multi-session CEBRA. **B.** Pairwise heatmaps over time bins sorted by behavioral angle, comparing behavioral differences (top left) and latent embedding distances of the two methods in **A**. The latent embedding distance is defined as the Euclidean distance between the latent embeddings with rows and columns ordered identically to the left panel. **C.** The alignment score for CANDY (blue) and multi-session CEBRA (purple). Solid lines: the average alignment score over 5 random seeds. Error bar: s.e.m. over 5 random seeds.

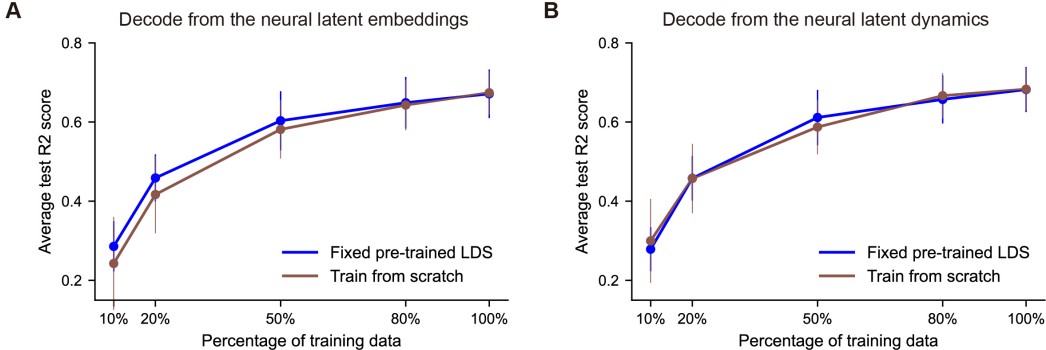

Figure S8: **Generalization of CANDY to held-out sessions in the monkey center-out reaching task. A.** Decoding performance on latent embeddings $\hat{u}$ across sessions as a function of the fraction of held-out session data used. The blue curve shows the results with the fixed pretrained LDS and behavior decoder from CANDY; the brown curve shows training from scratch on the held-out data. **B.** Same comparison for decoding from the inferred latent dynamics. Solid line: average performance over 2 held-out sessions, and each session's performance is averaged over 5 random seeds. Error bar: s.e.m. over 2 held-out sessions.

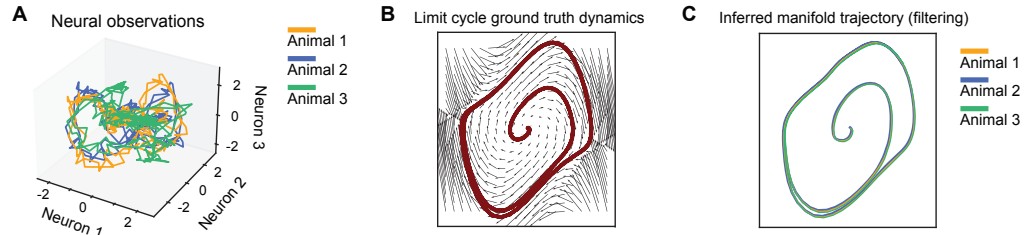

Figure S9: **Recovering shared latent factors trajectory from nonlinear dynamics. A.** Simulated neural observations of three animals, each generated by applying a distinct nonlinear mapping from the same underlying low-dimensional trajectory in (**B**). **B.** Ground-truth limit cycle dynamics trajectory. **C.** Latent trajectory inferred by CANDY for each animal with latent dimension set to four.

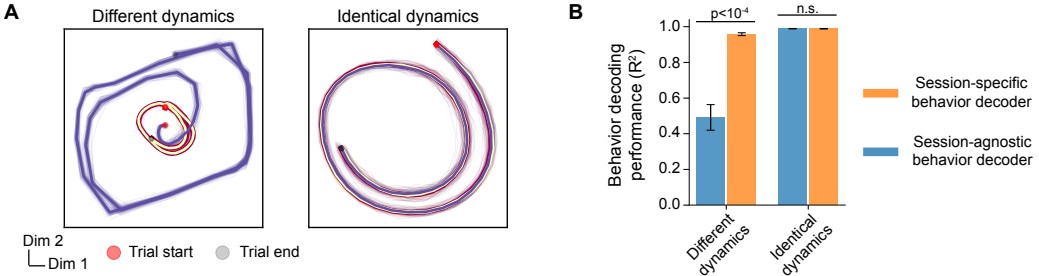

Figure S10: **Detecting divergent latent dynamics across sessions. A.** Latent factor trajectories inferred by CANDY in a synthetic dataset with three sessions. Left: two sessions were generated from the identical spiral dynamics, while the third followed a limit-cycle dynamics (Appendix A.3). Right: all three sessions were generated from the identical spiral dynamics (Appendix A.1). **B.** Behavior decoding accuracy of session-agnostic (Appendix C.2.1) and session-specific decoders (Appendix C.2.2). All tests are two-sided Wilcoxon signed-rank tests. Error bar: s.e.m over 20 seeds.

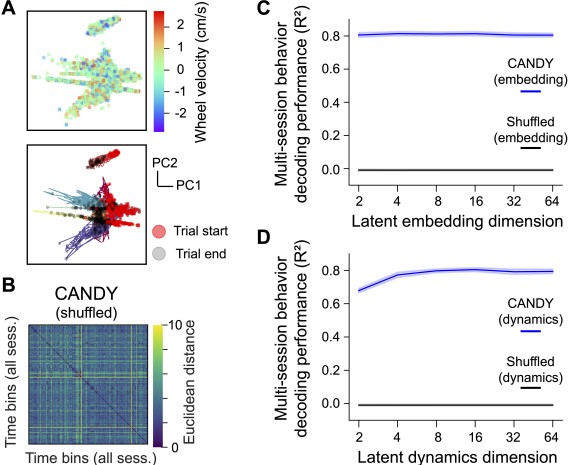

Figure S11: **Shuffled neural–behavioral control confirms that CANDY relies on genuine neural–behavioral relationships. A.** Same as Fig.3B but using the shuffled neural-behavior data. Top: colored by instantaneous wheel velocity. Bottom: colored by session IDs. **B.** Latent embedding distance of the shuffled data. The latent embedding distance is defined the same as Fig.3C. **C-D.** Multi-session behavior decoding performance ($R^2$) evaluated with a session-agnostic decoder (Appendix C.2.1) on (**C**) neural latent embeddings and (**D**) latent dynamics. Solid line: average over 8 sessions; for each session, shuffling performance is averaged over 20 random shuffles. Shaded area: s.e.m over 8 sessions.

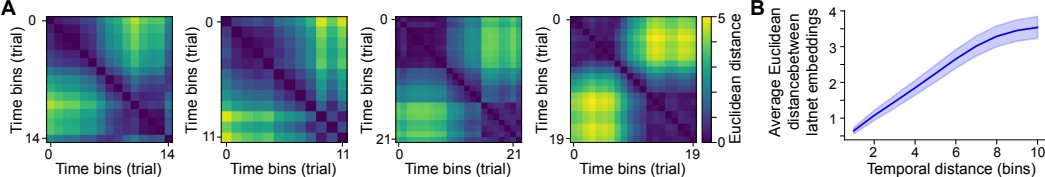

Figure S12: **Latent embeddings reflect temporal proximity within trials. A.** Example temporal similarity maps from four representative trials. Each heatmap is the pairwise Euclidean distance between embeddings at all time bins within the trial; embeddings from temporally adjacent bins remain close in latent space. **B.** Average Euclidean distance between embeddings as a function of temporal distance. Solid line: average over all trials and sessions. Shaded area: s.e.m. across all trials and sessions.

