# OpenReview forum: "Extracting task-relevant preserved dynamics from contrastive aligned neural recordings"
_NeurIPS.cc/2025/Conference — NeurIPS 2025 spotlight_

### Official Review · Reviewer_JGvB · 2025-07-01

**Clarity:** 3
**Significance:** 3
**Originality:** 3
**Rating:** 5
**Confidence:** 3

**Summary:**

Authors develop a method to generate a shared embedding space of neural recordings from many sessions to capture presumably low-dimensional preserved dynamics. They also fit a linear dynamical system to model the temporal evolution in the latent space derived from the created embedding. Their method uses contrastive learning methods to enforce that neural states with similar behavioral observations are mapped close by and vice versa. They apply their method to a synthetic, a mouse, and a monkey dataset and show that the learned embedding maintains similarity observed in the behavioural observations across sessions and subjects and allows the decoding of the observed behaviours.

**Questions:**

(1) The method encourages behavioral similarity by design, but it would be valuable to understand whether other meaningful properties are preserved or not. For example, does similarity between neural recordings persist in the embedding space? Is proximity in time or trials reflected?

(2) Regarding generalization, are you using the behavioral supervision loss here? If not, it’s unclear why the method would generalize. There doesn't seem to be anything explicitly enforcing that the embedding space is shared across sessions. Could you please clarify?

(3) The approach assumes that the decoded behavioral observations are task-relevant and can be reliably read from the neural data. An important control would be to shuffle the neural recordings relative to the behavioral data and show that under these conditions, the method cannot find a “good” embedding. Including such a control would help confirm that the embedding genuinely reflects neural-behavioral relationships.

(4) To better assess generalization, it would be helpful to test whether the model can map previously unseen neural states to the correct behavioral outputs. One suggestion would be to hold out specific trial types (e.g., certain movement angles or task periods) from training and include them only in the test set. Are these held-out examples mapped to the correct behavioral location in the embedding space? Can their associated behavior still be accurately decoded? This would help evaluate whether the embedding captures meaningful behavioral structure, even for novel data, or where its limitations are.

**Ethical Concerns:**

["NO or VERY MINOR ethics concerns only"]

**Final Justification:**

The authors present an interesting method for constructing shared latent embedding spaces across neural recording sessions using behavioural observations. The approach is tested on synthetic, mouse, and monkey datasets, demonstrating robustness and generalizability. They further compare their method to existing methods in a fair and helpful manner. I appreciate the authors’ thoughtful rebuttal, which directly addressed key concerns through additional experiments and clarifications. While some limitations to their method remain, these limitations are not critical, and the authors acknowledge these limitations clearly and outline directions for future work. Overall, I believe this is a strong and timely contribution to the field.

**Limitations:**

The approach is interesting and the results on preserving behavioral similarity are promising. However, it would be important to discuss whether preserving behavioral similarity is always desirable. In neuroscience, it's common for different cognitive processes to produce identical behavioral outputs - such as during evidence accumulation in integration tasks, where animals may show the same behavior while engaging in opposite internal processes. It would strengthen the paper to include a discussion of when and where this method is appropriate, and what the limitations might be in such cases. Alternatively (but probably outside the scope of this paper): How does the method perform on error trials? It would be interesting to see whether the model distinguishes between correct and incorrect trials in the embedding, and whether behaviorally similar but cognitively distinct trials are treated differently.

**Paper Formatting Concerns:**

no formating issues observed

**Quality:**

3

**Strengths And Weaknesses:**

It’s great that the method was tested across multiple datasets, which supports its robustness. Similarly, the comparison to other state-of-the-art methods is much appreciated, especially with the use of grid search to tune those methods for a fair evaluation. The analysis on the importance of different loss terms is interesting and adds depth to the understanding of the method.

The paper provides clear definitions of the loss components and the overall setup. However, it should be made clearer which specific loss terms were used to generate the results shown. This would help readers better interpret the findings and reproduce the experiments.

Incorporating an LDS (linear dynamical system) component in this method is a nice addition to improve interpretability. However, the paper doesn’t clearly demonstrate how this addition can be practically leveraged.

---

> ### Author Rebuttal · Authors · 2025-07-30
>
> Thank you for your clear summary of our work, and the constructive feedback. Please find below our point-by-point responses to your comments and questions. In all cases, we either added a new experiment or specific clarifications to the text to address your concerns. We look forward to your assessment!
>
> > Weakness 1: … it should be made clearer which specific loss terms were used to generate the results shown...
>
> Thank you for this suggestion. We used only the model loss and contrastive loss for Figures 2-3 and S1-S2. We added behavior supervision loss for Figures 5 and S3-S5. So, in total, we use two variations of the algorithm. We will make it clear in the paper by using a separate name for when the behavior supervision loss is used or not: CANDY and CANDY-sup.
>
> > Weakness 2: … the paper doesn’t clearly demonstrate how LDS addition can be practically leveraged.
>
> Thank you for making this point. Before discussing interpretability, we want to emphasize that the LDS component in CANDY is **not just an add-on**, but is central to the **scientific question** motivating this work: **can we extract a preserved latent dynamical system (not just representations!) underlying neural activity across different animals performing similar tasks?** While contrastive learning methods like CEBRA can yield behaviorally aligned embeddings, they do not model underlying temporal structure, and therefore cannot test whether shared latent dynamics exist across individuals. CANDY is designed to fill this gap. We chose LDS as a simple, interpretable, and computationally efficient way to model latent dynamics, and because LDSs are effective for local approximations of neural dynamics [1]. Figures 2 and 4 show that contrastive alignment is critical for recovering consistent dynamics, whereas behavioral supervision alone fails to do so, as indicated by weaker decoding from dynamics factors. Moreover, Section 4.2.4 and Figures 5 and S5 demonstrate that an LDS pretrained on multiple animals can be transferred to new animals without retraining. This provides compelling empirical evidence for the existence of shared latent dynamics across individuals.
>
> In terms of interpretability, Fig. 3 shows meaningful topology in the embedding space. In Fig. 3B, latent embeddings colored by wheel velocity reveal a smooth, 1D manifold consistent with task structure. We added the same analysis (will be placed in the Appendix) using LDS dynamical factors and observed similarly structured, low-dimensional geometry aligned with both wheel velocity and trial type. Moreover, we added an analysis on the relationship between the dynamical factors and behavioral variables across sessions. Specifically, we computed the correlation between each dimension of the LDS latent state and the observed wheel velocity across sessions when latent dimension = 8. We found dynamical factors of CANDY exhibited correlations $\approx$ 0.68±0.04 with the behavior, whereas that of DFINE and SLDS exhibited correlations $\approx$ 0.23±0.034 and 0.12±0.001.
>
> > Q1
>
> Good point! Beyond behavioral alignment (Figures 3B-C, S2, and S4), we find that CANDY’s latent embeddings also reflect meaningful temporal structure. We added an analysis of pairwise distances within trials and found that temporally adjacent time points remain close in latent space. Plotting Euclidean distance versus temporal separation shows clear temporal continuity in the embeddings, exactly what the LDS was designed to capture. We will add it to the Appendix.
>
> > Q2
>
> Yes! We used a behavior supervision loss in the generalization experiments, since  the pretrained sessions were unseen, so contrastive comparisons weren’t possible. In L257, we stated that both the LDS and behavior decoder from the pretrained model were frozen. To clarify, we will revise the sentence to “We froze the shared LDS and behavior decoder from the pretrained model and trained only the subject-specific encoder and neural decoder on a held-out third animal using both contrastive and behavior supervision losses (5 sessions).”
>
> > Q3
>
> Thanks for the suggestion! We performed this control and the results are as expected.
> We randomly shuffled behavioral data relative to neural activity 20 times to break their correspondence. Under these conditions, the model fails to converge—the training loss remains high, embeddings lack coherent structure, and behavior decoding drops to chance. Visualizations show no session alignment or interpretable organization, in contrast to Fig. 3B-C. These results confirm that CANDY’s ability to extract structured, behaviorally meaningful embeddings depends on genuine neural–behavioral relationships, not just model architecture. We will include these findings in the Appendix.
>
> > Q4
>
> Thanks for sharing this important suggestion! In the current manuscript, we tested whether the learned linear dynamical system and the behavior decoder can be generalized to unseen neural states from new subjects performing similar behaviors on both our mouse wheel-turning dataset (to be released if accepted) and a standard monkey center-out reaching dataset. Results show superior performance of the pretrained LDS and behavior decoder compared to training from scratch on the same sessions.
>
> To address your question, we tested whether the model can generalize to new behavioral states by holding out a movement direction in the monkey dataset. Unfortunately, both CANDY (test $R^2$: mean -0.1209, s.e.m. 0.0319) and CEBRA (test $R^2$: mean -0.2138, s.e.m. 0.0248) failed to generalize to the unseen behavior category, which will be added to the Appendix. This reflects the broader challenge of out-of-distribution generalization in machine learning. We will add the following to the Limitations and Outlook section: "While CANDY shows strong generalization of the pretrained LDS and behavior decoder to unseen sessions, both CANDY and CEBRA failed to generalize to unseen trial types (Appendix), which can be explored in future work."
>
> > Limitation 1: … discuss whether preserving behavioral similarity is always desirable… when and where this method is appropriate, and what the limitations ...
>
> Great point! Several recent studies have shown that different subjects can rely on similar neural dynamics within the same brain area to drive both well‑trained behaviors [2] and more naturalistic actions [3]. In our work, we likewise find that both mice in the wheel‑turning task and monkeys in the center‑out-reaching task exploit preserved dynamics. That said, it remains unclear whether such preserved dynamics generalize across all behavioral paradigms or brain regions, especially given that distinct internal processes can sometimes produce identical observable actions.
>
> To address this, we added a synthetic experiment where two sessions were generated from the same spiral dynamics, while the third used limit-cycle dynamics. All three shared the same linear behavioral readout. When trained on all three sessions, CANDY successfully aligned the two spiral sessions, while the third remained misaligned in the latent space. With a shared behavior decoder, the decoding performance is only ($R^2 \approx 0.9, 0.6, 0.1$ respectively), revealing a mismatch in underlying dynamics. Based on the misalignment observed in the latent space, we identified the divergent session and excluded it. Retraining on the two aligned sessions improved decoding back to $R^2\approx 0.99$. This result suggests that a drop in cross‑session decoding performance can serve as an initial indicator of when latent dynamics are not shared.
>
> We will make explicit in the Limitation and outlook section by adding: “While CANDY can facilitate the identification of distinct underlying dynamics as shown in the synthetic experiment, developing methods to distinguish truly distinct neural processes that yield similar behaviors remains an important direction for future work.”
>
> > Limitation 2: … How does the method perform on error trials? … whether the model distinguishes correct and incorrect trials … behaviorally similar but cognitively distinct trials are treated differently.
>
> Thank you for this insightful suggestion. As our primary goal in this manuscript is to demonstrate CANDY’s ability to recover preserved, behaviorally relevant dynamics, we have not yet evaluated its performance on error trials. Nonetheless, we agree that probing whether CANDY embeddings can distinguish correct from incorrect trials, especially when the overt behavior remains similar, would be highly informative. Due to time constraints, we will reserve the full analysis of correct versus error trial dynamics for the period leading up to the camera‑ready deadline, if accepted, and include it in the appendix.
>
>
> We appreciate your thoughtful and challenging questions, which have helped us strengthen both the conceptual and empirical aspects of our work.  We hope that our explanation and the newly conducted experiments would address your concerns. We look forward to your reassessment!
>
> [1] Abbaspourazad, Hamidreza, et al. "Dynamical flexible inference of nonlinear latent factors and structures in neural population activity." Nature Biomedical Engineering 8.1 (2024): 85-108.
> [2] Safaie, Mostafa, et al. "Preserved neural dynamics across animals performing similar behaviour." Nature 623.7988 (2023): 765-771.
> [3] Nair, Aditya, et al. "An approximate line attractor in the hypothalamus encodes an aggressive state." Cell 186.1 (2023): 178-193.

---

> > ### Comment · Reviewer_JGvB · 2025-08-03
> >
> > Thank you for the clarification and the additional work done by the authors. I appreciate the extra experiments and the clear reporting of the model's strengths and limitations. I am happy to raise my overall score to 5: Accept, as I believe the presented work is a promising contribution to the existing work on this highly relevant topic, with well-designed experiments that clarify the respective importance and function of the method’s added components.

---

> > > ### Author Response · Authors · 2025-08-03
> > >
> > > We are very grateful for your thoughtful feedback and for raising your score to 5: Accept! Your recognition of our contribution, the original and the additional experiments, and the report of both strengths and limitations is especially encouraging! Your suggestions have been significantly helpful in strengthening both the conceptual framing and empirical validation of our approach. We’re delighted that our analyses helped clarify the roles of the method’s components, and we truly appreciate your support and the time you put into the review process!

---

### Official Review · Reviewer_P5Sz · 2025-07-01

**Clarity:** 3
**Significance:** 3
**Originality:** 2
**Rating:** 4
**Confidence:** 4

**Summary:**

This work proposed CANDY, and end-to-end framework that aligns neural and behavior data using rank-based contrastive learning, and it also integrate with a linear dynamical system to capture the temporal associations. The optimization include a contrastive loss to align the neural latent embeddings with the continuous behavior samples, a behavior decoding loss, as well a linear dynamical system with parameters to be jointly optimized. It contributes to increase the generalizations across data, which allows learning the embeddings across sessions and subjects (also include unseen sessions), and demonstrates superior performance compared to existing baselines.

**Questions:**

1. In Figure 3b, it is unclear how CANDY is superior compared to CEBRA?
2. What is the computational cost and latency of the proposed method compared to existing approaches?
3. Are the parameters estimated in the linear dynamical system eqn (4) and eqn (5) identifiable and interpretable?

**Ethical Concerns:**

["NO or VERY MINOR ethics concerns only"]

**Final Justification:**

Thanks for the detailed responses from the authors, especially on clarification on using a LDS model, while still could introduce nonlinearity during the modeling. And adding comparisons with existing baseline Stitch-LFADS LFADS, and demonstrates superior performance. Those efforts have addressed most of my concerns, I would like to improve my score to borderline accept as positive score. Thanks for your efforts!

**Limitations:**

1. Lack of evaluations and comparisons with other SOTA baselines discussed in the related works [34-39].
2. The assumption of linear dynamical system (LDS) as a prior in the model might not be sufficient to capture the complexity of the neural dynamical systems,  and generalizable to more complex situations.

**Paper Formatting Concerns:**

Not applied.

**Quality:**

3

**Strengths And Weaknesses:**

**Strengths**
1. This method contribute to an important question on representation learning from neural activities for behavior decoding, and aim to improve the generalization of the method across subjects and sessions.
2. Compared to existing approaches, this model focused on using contrastive loss aligning with continuous behaviors, and incorporated with a dynamical constraint to enable temporal consistency, which is superior than existing baselines such as CEBRA.
3. This methods are evaluated on multiple datasets from mouse to macaque, demonstrate its generalizations across animal species.
4. It demonstrates superior performance on decoding performance and its performance are less sensitive to choice of hyperparameters such as latent dimension.

**Weakness**
1. Explain the choice of linear dynamical system, discuss whether it would be sufficient to capture the nonlinearity and nonstationarity of the dynamical system. It would be valuable to introduce ablation study on the complexity of dynamical system.
2.  Lack of comparisons with other baseline for behavior decoding such as stich-LFADS, POYOs, NDT-2, etc. [34-39] as discussed in the related works.

---

> ### Author Rebuttal · Authors · 2025-07-30
>
> We thank the reviewer for their constructive feedback throughout their review. Below, we provide point-by-point responses to your questions and comments. In all cases, we either added a new experiment or clarification to the manuscript.
>
> To begin with, we’d like to address a few misunderstood contributions in the summary, which we believe is the main reason behind confusions later in the review.
>
> *increase the generalizations across data*: Our goal is not to improve behavior decoding generalization across datasets. Instead, we started with a biological question: **how to extract the task-relevant preserved latent dynamics**, motivated by evidence of shared strategy in certain brain regions shared across sessions while doing similar tasks [1]. We used CANDY as a framework for investigation and demonstrated its effectiveness on two datasets: a new 2p calcium imaging dataset on mouse wheel-turning task collected by ourselves (will be released upon acceptance) and a public monkey electrophysiology dataset.
>
> *demonstrates superior performance compared to existing baselines*: We are not aiming to improve cross-session behavior decoding performance. Instead, we treat behavior decoding from neural embeddings and dynamic factors as a **diagnostic** way to **evaluate alignment quality and whether a preserved dynamics is correctly captured**. If our model matches or exceeds benchmark performance despite biologically motivated constraints, it suggests the preserved dynamics are well learned. Surprisingly, our model shows superior performance, further supporting both the existence of preserved dynamics and our model’s effectiveness.
>
> To emphasize these distinctions, we will revise the contributions at the end of the introduction as:
>
> “Our key contributions include:
> 1) Methodologically, we develop a new framework to extract preserved dynamics underlying continuous behavior across sessions and animals using a rank-based contrastive objective and a linear dynamical system.
> 2) Conceptually, we show that contrastive alignment is essential for uncovering preserved dynamical systems, not merely a decoding aid.
> 3) Biologically, while prior work shows preserved neural motifs across animals [1], none have tested if these can arise from the same dynamical system. We show that a dynamical model trained on one set of animals transfers to held-out individuals, indicating the existence of preserved dynamical structures in a specific brain region under a given task.
> 4) Experimentally, we collected a new cross-day multi-animal 2p calcium dataset and will release it upon acceptance.”
>
> > Weakness1 & Limitation2
>
> Our primary goal is to build a model that **extracts preserved, interpretable latent dynamics shared across sessions**. Beyond its clear interpretability, CANDY leverages learnable nonlinear encoders to grant the latent space far greater flexibility than a standard LDS would allow [2]. Intuitively, if we map neural activities to latent variables via $z = f(x)$ and impose an LDS prior $\dot z = A z$, the induced dynamics on the observations become $\dot x = (\nabla f(x))^{-1} A\,f(x)$, where $\nabla f(x)$ is the Jacobian of $f$. Since $f(x)$ is learnable and nonlinear, the resulting latent variables induce nonlinear dynamics on the observed neural activity.
>
> “While we focus here with an LDS prior to test our core hypothesis of the existence of preserved dynamics, whether replacing the LDS with different types of nonlinear dynamics in the CANDY framework can be beneficial is an interesting future direction.” We will add this to the Discussion section.
>
> To address this concern empirically, we performed a new synthetic limit cycle experiment, involving *nonlinear*, *nonstationary* dynamics. Despite this complexity, CANDY successfully aligns the sessions and recovers the shared 2D limit cycle in the latent space when emphasizing on the contrastive loss (similar as Fig. 2C). This shows that the combination of a flexible nonlinear encoder with a linear dynamics prior is sufficient to recover complex underlying dynamics. We will add this to Fig. 2.
>
> > Weakness2 & Limitation1
>
> We believe there is a misunderstanding about the primary motivation of our work and we apologize if the information in the Related work is misleading.
>
> While benchmarking large-scale models on behavior decoding is important, CANDY is not designed to focus on maximizing behavior decoding performance, as we said above. We will add “These transformer-based architectures focus primarily on behavior decoding, whereas CANDY focuses on extracting interpretable preserved dynamics across sessions.” to the Related work.
>
> It is a great suggestion to compare with Stitch-LFADS since it also models a shared dynamical structure. We grid-searched 800 hyperparameters for both datasets independently. The mean±s.e.m of testing $R^2$ are (table entry: using a universal behavior decoder \ session-specific behavior decoder):
>
> Mouse dataset
> |Latent dim|Stitch-LFADS|CANDY|
> |-|-|-|
> |2|-0.0093±0.0051 \ 0.5380±0.0340|0.6787±0.0130 \ 0.6798±0.0133|
> |4|0.0260±0.0839 \ 0.6267±0.0195|0.7727±0.0157 \ 0.7859±0.0146|
> |8|0.0315± 0.0291 \ 0.6910±0.0151|0.7976±0.0111 \ 0.8062±0.0136|
> |16|0.0734±0.0556 \ 0.7223±0.0131|0.8046±0.0119 \ 0.8108±0.0125|
> |32|0.1504±0.0836 \ 0.7548±0.0116|0.7917±0.0159 \ 0.8005±0.0135|
> |64|0.2704±0.0758 \ 0.7743±0.0136|0.7937±0.0134 \ 0.8023±0.0124|
>
> Monkey dataset
> |Latent dim|Stitch-LFADS|CANDY|
> |-|-|-|
> |2|-0.0136±0.0044 \ 0.0920±0.0075|0.6453±0.0107 \ 0.6178±0.0118|
> |4|-0.0078±0.0056 \ 0.2185±0.0106|0.6089±0.0130 \ 0.6158±0.0119|
> |8|-0.0001±0.0080 \ 0.3822±0.0108|0.6015±0.0126 \ 0.6106±0.0122|
> |16|-0.0020±0.0089 \ 0.5041±0.0096|0.5714±0.0134 \ 0.6095±0.0122|
>
> These results indicate that stitch-LFADS failed to align the dynamics across sessions.
>
> > Q1
>
> This figure is not meant to show CANDY’s superiority over CEBRA, but to highlight the advantage of those trained with contrastive learning over those without. Specifically, Fig. 3B shows that contrastive methods (CANDY and CEBRA) can align neural latent embeddings across sessions using behavior as an anchor, a property not observed in DFINE, which serves as a baseline without contrastive alignment.
>
> Direct comparisons to CEBRA are shown in Figures 3C-D, S2, S3B and S4. CANDY produces latent embeddings with significantly stronger correlations to ground-truth behavior and higher behavior decoding performance, indicating **more interpretable** and **behaviorally meaningful** representations.
>
> > Q2
>
> Due to limited access to high-performance GPUs, all experiments were run on CPU. On an AMD Ryzen 9 9950X, training CANDY on 8 sessions (~250 neurons, ~30 time steps each, batch size 2048) takes about 10 hours. For comparison, Stitch-LFADS takes ~8 hours and CEBRA ~6 hours under similar settings. With a high-VRAM GPU, we expect CANDY’s training could be accelerated by an order of magnitude or more.
>
> > Q3
>
> Theoretically, both are important questions of current research. Empirically, the answer is yes.
>
> Identifiability: Building on recent work on identifiable VAEs, CEBRA proves that latent variables learned with categorical contrastive loss are linearly identifiable. We apply a contrastive loss suited for regression, while preserving the same structure. We additionally fit a linear dynamical system to model these embeddings and LDS parameters are identifiable up to orthogonal transformations. Therefore, though there is still a gap in the broader literature regarding the identifiability in the case of joint fitting of contrastive objectives and their dynamics, our work builds on existing identifiability proofs in terms of the distinct parts of the model. We will include a discussion of this in the manuscript.
>
> Interpretability: While LDS identifiability means that the exact matrices (e.g., dynamics matrix A) are not uniquely determined, the topology and temporal evolution of the latent trajectories are still meaningful in a task-aligned context, which is precisely our contribution. We aligned the latent variables using the contrastive loss and continuous behavioral signals, and illustrate their interpretability with examples:
>
> - In Fig. 3B, neural latent embeddings colored by wheel velocity reveal a smooth 1D manifold aligned with task structure. To further highlight this, we will update the figure with an additional row colored by trial type (leftward vs. rightward cursor onset), where the two wings of the trajectory correspond cleanly to the two trial conditions, reinforcing the interpretability of the learned latent geometry.
> - In Fig. 3C, the distance matrix of CANDY’s latent embeddings (top right) shows a block-diagonal structure that matches the behavioral similarity matrix (top left), indicating that the latent space faithfully encodes behavioral structure.
> - The behavioral interpretability is also evident in the center-out reaching task (Fig. S4A), where the latent representations clearly organize the eight reach directions into distinct clusters.
>
> In summary, while LDS and encoder parameters are expected to be linearly identifiable, empirically, the learned latent trajectories are behaviorally aligned, geometrically structured, and interpretable, supporting CANDY’s goal of uncovering shared task-relevant dynamics across sessions. In that regard, our work provides an important empirical contribution to the literature. We hope these clarify your question.
>
> We hope our clarifications on the motivations and contributions of the paper and the choice of LDS address your concerns. If there are any other points we can clarify or further experiments we can conduct to strengthen your assessment, please let us know.
>
> [1] Safaie, Mostafa, et al. "Preserved neural dynamics across animals performing similar behaviour." Nature 623.7988 (2023): 765-771.
> [2] Abbaspourazad, Hamidreza, et al. "Dynamical flexible inference of nonlinear latent factors and structures in neural population activity." Nature Biomedical Engineering 8.1 (2024): 85-108.

---

> ### Author Response · Authors · 2025-08-07
>
> Dear Reviewer, we hope our explanations on the contribution of our work and interpretation of the LDS, additional experiments on stitch-LFADS and nonlinear dynamical systems, and the modification to the Contribution, Related work, and Discussion section address your concerns. Looking forward to your reassessment!

---

### Official Review · Reviewer_oySm · 2025-07-02

**Clarity:** 3
**Significance:** 3
**Originality:** 3
**Rating:** 6
**Confidence:** 4

**Summary:**

The authors propose CANDY, a dimensionality reduction method for neural data that fits a shared LDS over multiple sessions with aligned embeddings. A key step is using a contrastive loss to align embeddings  based on the continuous behavioural label associated with each time point. This encourages the learned embeddings to group together neural data points with similar behaviours, even across sessions. They apply the method to one ground truth simulation, and a couple datasets. One interesting finding is that the contrastive loss is necessary to find behaviourally relevant embeddings, and simply using a supervised behavioural loss is not sufficient.

**Questions:**

- Line 112, definition of S_h,j, I think that should be \leq, not \geq. Is that a typo?
- The ground-truth analyses (section 4.1) should include examples of where the model breaks down. For example, what happens if the behavioural readout is highly nonlinear, or worse yet if the mapping varies from session to session?
- Lines 260-266: Can the authors give more details here? Is the model trained from scratch only trained on a single session?
- In neuroscience experiments, it is sometimes unclear if different animals are using the same dynamical strategies to perform a task. Is the model robust to different latent dynamics across sessions? Can it be used to discard sessions that are deemed too dynamically different from the other sessions?

**Ethical Concerns:**

["NO or VERY MINOR ethics concerns only"]

**Final Justification:**

The authors have made a strong case for CANDY, including promising results in comparison to stitch-LFADS, and helpful clarifications about the difference between their model and competitors. I think the model hits the perfect balance between simplicity and expressivity to make an impact in neuroscience. Inferring latent dynamics across sessions is a very important question in neuroscience, and this is one of the few methods I've seen that tries to do this while maintaining an interpretable architecture. The authors therefore have my full support for this paper.

**Limitations:**

Somewhat. The authors could be clearer about the limitations of their model and when they expect it to fail, for example in out-of-sample data where there may be session-specific nonlinearities.

**Quality:**

3

**Strengths And Weaknesses:**

Strengths:

This is a really nice paper with a cool idea to align latents using their behavioral labels by leveraging contrastive learning.The claims are generally well supported, and the analyses of the contribution of the contrastive loss are interesting.

As far as I know the model is a novel application of an existing rank-based contrastive loss to a new problem regarding alignment of neural dynamics (although I am not an expert on contrastive learning so I am not 100% sure on the novelty). It is certainly a step forward compared to CEBRA or DFINE. I believe the model will be broadly interesting to NeurIPS and the analyses of the contrastive loss offers some insights regarding the important considerations for aligning latents. I can certainly see future researchers building upon this idea for new alignment algorithms.

It's well written and clear for the most part, which I appreciated as a reviewer.

Weaknesses:

Although I really like this submission and find it to be high quality overall, there are two fundamental points that really should be improved.
- The method is demonstrated in two neuroscience data sets, although one dataset receives much more attention in the main text. But the paper is missing ground truth experiments on out-of-sample data.
- The discussion is very short and should be expanded to include more limitations and relationship to other work on latent dynamics inference and contrastive learning in neuroscience.

---

> ### Author Rebuttal · Authors · 2025-07-30
>
> Thank you for this summary, which hits the mark perfectly. We appreciate your support of our manuscript and the constructive feedback throughout the review. Please find below our point-by-point responses to your questions and comments.
>
> > Weakness 1: … one dataset receives much more attention in the main text.
>
> An important reason for this is because we collected this dataset ourselves specifically to evaluate generalization in a setting where we had full control over preprocessing and task design. Nonetheless, we show applications to the public monkey dataset, as is the convention in the field currently. We believe that the use and introduction of **a new, cross-subject calcium imaging dataset**, which will be made publicly available upon acceptance, is a strength of our study and part of the novelty. We hope it provides value to the community by expanding the types of datasets used to evaluate neural dynamics models.
>
> > … missing ground truth experiments on out-of-sample data.
>
> We acknowledge this limitation, but emphasize that we showed in Figures 5 and S5 that the LDS and the decoder can be generalized to a held-out animal. To prove this on a dataset with ground truth, we added an additional experiment on the synthetic dataset by helding-out one session that shared the same spiral latent dynamics but different neural observations readout and noise perturbation. By fixing the LDS and tuning the encoder parameters only for the held-out session, we observe that the model can correctly infer the spiral latent trajectory on the held-out dataset, following the pretrained LDS flow field. We will add this result to the revised manuscript.
>
> > Weakness 2: The discussion … expanded to include more limitations and relationship to other work on latent dynamics inference and contrastive learning ...
> Limitations: … clearer about the limitations of their model and when they expect it to fail, … session-specific nonlinearities.
>
> We agree with this assessment. In the initial submission, due to page limit, we had to shorten our discussion section. In the revised manuscript, we will include more work from the broader neuroscience literature that employs contrastive learning:
>
> “**Contrastive learning in neuroscience.** Recent studies have explored contrastive learning for extracting meaningful neural latent representations. CEBRA [1] embeds neural activity with noncausal convolutional neural network by using contrastive objective on temporal similarity and discretized behavioral similarity.  MARBLE [2] introduces a geometric deep learning framework that embeds local flow fields computed from neural activities into a latent space using contrastive objectives. However, it requires *post hoc* alignment for downstream behavior decoding across sessions. While both methods produce useful low-dimensional representations, they do not explicitly model shared latent dynamics across sessions or individuals.”
>
> and add the following to an explicit **Limitations and outlook** section which will be included in the manuscript:
>
> “CANDY offers a principled approach for aligning population activity and extracting shared latent dynamics across sessions and subjects, but it has limitations. First, it relies on a linear dynamical system (LDS), which may be too restrictive for highly nonlinear neural dynamics. While Koopman theory suggests that higher-dimensional LDSs can approximate complex systems, we lost the ability to truly reveal the dynamical structure and the latent dimension. Second, CANDY assumes a shared linear behavioral readout across subjects, which may overlook preserved dynamics with nonlinear behavioral readout or subject-specific readouts. Although we found this assumption sufficient in our experiments, incorporating distinct behavioral readouts is a promising direction for future work.”
>
> > Q1: Line 112, definition of S_h,j, I think that should be \leq, not \geq. Is that a typo?
>
> No, it should be \geq. Please allow us to elaborate: $S\_{h,j}$ is defined as a set of variables whose distance to anchor $\hat{u}_h$ is larger than the distance between anchor $\hat{u}_h$ and $\hat{u}_j$, and hence are less similar to $\hat{u}_h$ than $\hat{u}_j$. Therefore in the contrastive loss, we pull $\hat{u}_j$ closer to $\hat{u}_h$ compared to other variables in $S\_{h,j}$.
>
> > Q2: The ground-truth analyses … where the model breaks down. … if the behavioral readout is highly nonlinear, or … the mapping varies from session to session?
>
> Great question! In the model we presented, we always set the behavior decoder to be linear. To address your question, we added a few more synthetic experiments:
> - Same spiral dynamics and same nonlinear behavior readout
> - Same spiral dynamics but different linear readouts for each session
>
> We observe that CANDY breaks in both situations. We believe this is an interesting future direction. One potential approach is that contrastive learning cannot just take in the behavior label naively and a behavior encoder might be needed for such a situation for recovering the preserved dynamics. We will add this to the Limitations and outlook section as written in the above response.
>
> > Q3: Lines 260-266: ... Is the model trained from scratch only trained on a single session?
>
> We apologize for the confusion. No, the model trained from scratch was trained on all held-out sessions. We will change L263 from “model trained from scratch on the same sessions” to “model trained from scratch on all held-out sessions”.
>
> > Q4: ... Is the model robust to different latent dynamics across sessions? Can it be used to discard sessions that are deemed too dynamically different from the other sessions?
>
> Great question! To explore if the model can distinguish sessions with different dynamics, we added a synthetic experiment where two sessions were generated from the same spiral dynamics, while the third used limit-cycle dynamics. All three shared the same linear behavioral readout.
>
> When trained on all three sessions, CANDY successfully aligned the two spiral sessions, while the third remained misaligned in the latent space. The decoding $R^2$ value for each dataset is only $\approx 0.9, 0.6, 0.1$ respectively, revealing a mismatch in the underlying dynamics. Based on the misalignment observed in the latent space, we identified the divergent session and excluded it. Retraining on the two aligned sessions improved decoding back to $R^2\approx 0.99$.
>
> This result suggests that CANDY can not only recover shared latent dynamics but potentially also serve as a diagnostic tool to flag sessions or subjects that deviate in their dynamical strategies. We see this as an exciting direction for extending this work. We will include the new synthetic experiment in the Appendix and add the following to the above Limitation and outlook section: “Third, while CANDY could facilitate the identification of distinct underlying dynamics as shown in the synthetic experiment, distinguish different neural processes that yield similar behaviors is an interesting extension to this work.”
>
>
> We appreciate your thoughtful and constructive questions, which have helped us strengthen both the conceptual and empirical aspects of our work. We hope that our explanation and the newly conducted experiments would address your concerns. We look forward to your reassessment!
>
> [1] Schneider, Steffen, Jin Hwa Lee, and Mackenzie Weygandt Mathis. "Learnable latent embeddings for joint behavioural and neural analysis." Nature 617.7960 (2023): 360-368.
> [2] Gosztolai, Adam, et al. "MARBLE: interpretable representations of neural population dynamics using geometric deep learning." Nature Methods (2025): 1-9.

---

> > ### Comment · Reviewer_oySm · 2025-08-03
> >
> > Thank you for your detailed replies to all reviewers. I believe the results are quite convincing, and with the simplicity of the model I think it can have a significant impact at least in neuroscience. Therefore I will raise my score.
> >
> > Small point: Regarding Q1, I was confused by the phrase "higher ranks" in line 111 which is ambiguous (so I interpreted S_h,j as a neighborhood). It could be useful to clarify this for future readers, but I leave it to the authors to decide.

---

> ### Author Response · Authors · 2025-08-03
>
> Great point! We apologize for the confusion in the text and will add the following sentences (bold) to the paragraphs in L110-115.
>
> Given an anchor $\hat{u}_h$ with the associated behavior $y_h$, for any other embedding $\hat{u}_j$, we define a set of embeddings that have higher *ranks* than $\hat{u}\_j$ in terms of the behavioral label distance with respect to the anchor as $S\_{h, j} := \\{\hat{u}\_k \mid k\neq h, d(y\_h, y\_k)\geq d(y\_h, y\_j)\\}$, where $d(\cdot, \cdot)$ is the $L_1$ distance between the two behaviors. **In other words,** $S\_{h,j}$ **is defined as a set of embeddings whose distance to the anchor**  $\hat{u}_h$  **is *larger* than the distance between the anchor**  $\hat{u}_h$ **and** $\hat{u}_j$ **, and hence are *less* similar to the anchor**  $\hat{u}_h$ **than** $\hat{u}_j$**.** Define the "likelihood'' of $\hat{u}_j$ given the anchor and the higher rank set as:
>
> Eq. (1)
>
> where $\text{sim}(\cdot, \cdot)$ is the similarity measure between two latent embeddings (*e.g.*, negative $L_2$ norm) and $\tau$ is the temperature parameter.  **Here, the likelihood will be maximized to push** $u_j$ **closer to the anchor** $u_h$ **in the embedding space than any other embeddings in the rank set.**
>
> We sincerely thank the reviewer for the thoughtful engagement with our work and for the updated score. We’re especially grateful that you found the results convincing and recognized the potential impact of our model in neuroscience. Your suggestions have significantly helped us strengthen both the conceptual framing and empirical validation of our work. We truly appreciate your support and the time you put into the review process!

---

### Official Review · Reviewer_6wRv · 2025-07-03

**Clarity:** 3
**Significance:** 3
**Originality:** 2
**Rating:** 4
**Confidence:** 4

**Summary:**

The authors propose CANDY, a contrastive learning approach to align latent encodings of neural activity across multiple sessions and subjects by using continuous behavioral variables as a shared anchor and per-session neural encoders. In addition to the contrastive objective, there is an optional behavior prediction objective and a k-step ahead neural prediction loss obtained by additionally fitting a linear dynamical system to the latent space and a nonlinear latent-to-neural decoder. The method is validated using a mouse wheel-turning task with calcium imaging and a monkey center-out reaching task using electrophysiological data. CANDY produces latent spaces that successfully predict behavior and generalize across recording sessions and individuals. Additionally, the latent distance matrices (across timebins) better reflect behavioral distances than other methods (CEBRA and DFINE). Lastly, in ablation experiments, the authors show the contrastive loss is necessary for the good performance they observe.

**Questions:**

Do you have evidence that the LDS improves the model? (See last paragraph of "Strengths and Weaknesses")

Will the wheel turning data be made public upon publication? It seems like it would be a useful resource.

Minor Comments
* Figure 4a and 4b should plot the contrastive-but-not-behavioral embeddings and matrices from Figure 3 for easy comparison.

**Ethical Concerns:**

["NO or VERY MINOR ethics concerns only"]

**Final Justification:**

* The paper explores behavioral-aligned contrastive learning for neural data with latent dynamics, a very promising and interesting research direction.
* The model performs well in terms of behavioral predictions, and is able to align data from multiple individuals and recording sessions on two real-world datasets.
* Resolved in discussion period: the learned dynamics are informative
* Could be strengthened: evidence of behaviorally relevant shared neural dynamics across subjects. The evidence presented in Figure 5 is, in my view, fairly weak.

**Limitations:**

Training a new encoder for each session could be cumbersome and may not work as well for sessions with neural but not behavioral data.

**Quality:**

3

**Strengths And Weaknesses:**

Contrastive learning for neural/behavioral modeling is a promising research direction with several encouraging recent results (CEBRA, MARBLE, etc.).

The various components of the method are well-justified: the session-specific encoders, the  behaviorally-anchored neural embeddings, the neural decoders, and the linear dynamical system. However, these components are all fairly standard, so I consider CANDY to have modest novelty.

The behavioral prediction results (Figures 3d-e, S3b-c) seem to be very good compared to baseline models. Additionally, the neural/behavioral alignment as visualized in Figures 3c and S4b is very impressive. Generally, the experiments seem to be high quality.

MARBLE (Gosztolai et al. 2025, Nature Methods), as a contrastive learning method for neuroscience applications, is very relevant and should be cited. BLEND (Guo et al. 2025, ICLR) is also shares similar aims to CANDY. Additionally, there should be a discussion of the specific ways in which MARBLE and CEBRA differ from CANDY. Contrastive learning in neuroscience could easily fill a whole paragraph in the related works section.

Is the LDS useful at all? It does not seem to help predictive performance to use the filtered latents (e.g. Figure 3 d vs. e). But it may still help the model learn a good latent space during training. I view the lack of evidence for the usefulness of the LDS as a major weakness of the paper, and would consider raising my score if can be addressed.

---

> ### Author Rebuttal · Authors · 2025-07-30
>
> Thank you for the summary and your constructive feedback. Please find our point-by-point responses to your comments and questions below. In all cases, we either performed new experiments or added clarification to the text to address your concerns. We look forward to hearing your assessment.
>
> > Weakness 1: The various components of the method are well-justified: … so I consider CANDY to have modest novelty.
>
> While these are indeed important building blocks of our work, we respectfully note that CANDY’s contributions go well beyond simply combining standard modules:
>
> First, our **methodological** advance lies in *developing a new framework to extract preserved dynamics underlying continuous behavior across sessions and animals using a rank-based contrastive objective and a linear dynamical system*. This shows two specific novelties compared to previous methods: (1) this goes beyond the limitation of other behaviorally-anchored neural embedding methods, such as CEBRA, which have to discretize behaviors before applying the contrastive objective; (2) the behaviorally-anchored neural embeddings are also constrained by a linear dynamical system, which assists the learning of the preserved dynamics by maintaining the temporal structure of the embeddings within a trial.
>
> Second, as a **conceptual** novelty, we show that *contrastive alignment is essential for uncovering preserved dynamical systems*, not merely a decoding aid. This is especially important for neuroscience as neural recordings show different degrees of heterogeneity, including number of neurons and the exact location of the recording. In Figures 2 and 4, removing the contrastive term causes the latent trajectories and the LDS to collapse. This highlights a new role of contrastive learning: as an alignment tool for uncovering preserved dynamical-structure across sessions, rather than just representation learning as in CEBRA.
>
> Third, our work addresses an important **biological** question: while earlier work has shown that neural activity motifs are preserved across animals performing the same tasks [1], no work to date has investigated whether these motifs can be supported by the same dynamical system equations. Here, we show that an LDS trained on one set of animals transfers directly to held-out individuals, *indicating the existence of preserved dynamical structures in a specific brain region during a specific task*. In Figures 5 and S5, we froze both the learned LDS and the behavior decoder and still achieved high decoding performance on unseen animals/sessions. This demonstrates that the latent dynamics inferred by CANDY are not only interpretable, but also consistently preserved across subjects in the mouse striatum during the wheel-turning task and in the monkey motor cortex during the reaching-out task. This is important because this shows a unique cross-subject generalization of dynamical models of the animal brains.
>
> Finally, on the **experimental** side, we ***designed and collected a new 2p calcium cross-animals dataset ourselves***, which includes 3 animals with 5 sessions each (15 sessions total), and each session contains ~200 trials with recordings from ~200 neurons in the striatum. We will make the dataset available to the community if accepted.
>
> We hope these clarifications would address your concerns of the originality of this work.
>
> > Weakness 2: MARBLE … should be cited. … discussion of the specific ways in which MARBLE and CEBRA differ from CANDY… the related works section.
>
> Thank you for referencing these papers. We agree that contrastive learning is a rapidly evolving area in neuroscience and appreciate the opportunity to better position our method in this context. We will add the following paragraph to the Related Work section:
>
> “**Contrastive learning in neuroscience**. Recent studies have explored contrastive learning for extracting meaningful neural latent representations. CEBRA [2] embeds neural activity with noncausal convolutional neural network by using contrastive objective on temporal similarity and discretized behavioral similarity.  MARBLE [3] introduces a geometric deep learning framework that embeds local flow fields computed from neural activities into a latent space using contrastive objectives. However, it requires *post hoc* alignment for downstream behavior decoding across sessions. While both methods produce useful low-dimensional representations, they do not explicitly model shared latent dynamics across sessions or individuals.”
>
> > Weakness 2: BLEND also shares similar aims to CANDY.
>
> BLEND develops a model-agnostic paradigm to distill behaviorally relevant neural latent embeddings from an existing model for use on new datasets lacking behavior signals. In contrast, CANDY develops a framework to extract preserved dynamics across animals by using behavior as an anchor to align heterogeneous neural populations (e.g., number of neurons and location of the recordings). We will add the following sentences to the Related work section:
> “The BLEND framework [4] uses a teacher-student setup to distill behaviorally relevant information to student networks when there is a lack of behavior data in a new dataset, but does not focus on extracting shared task-relevant dynamics across animals.”
>
> > Weakness 3 & Q1: Is the LDS useful at all? …
>
> This is a critical aspect of our work and you are correct that the LDS is not designed to improve behavior decoding performance. Below, we highlight two distinct reasons to address why LDS is useful:
>
> First, the LDS module is essential for addressing an important scientific question: **Can we extract a preserved and latent dynamical system (not just representations!) underlying neural activity across animals performing similar tasks?** While contrastive learning methods like CEBRA can produce behaviorally meaningful embeddings, they do not model the underlying dynamics and thus cannot determine whether shared dynamical motifs exist across subjects. This is precisely what CANDY was designed to address. More importantly, in Section 4.2.4, Figures 5 and S5, we demonstrate that a pretrained LDS learned from several animals generalizes to new animals with high decoding performance, without retraining the LDS. This provides **biological evidence** that the **neural activities in the same brain region can exhibit shared dynamics across individuals in similar well-trained behaviors**.
>
> Second, LDS is a key component because our methodological novelty lies in **extracting a preserved, interpretable dynamical system via behaviorally aligned neural embeddings, not in boosting behavior decoding performance using LDS**. As shown Figures 2 and 4, omitting the alignment step prevents recovery of a coherent LDS. Fig. 4 further illustrates that **contrastive objective is critical for uncovering preserved dynamics**: replacing contrastive loss with behavior supervision achieves similar decoding from embeddings but *fails* to recover meaningful dynamics, as seen in much worse decoding from the latent dynamical factors. To our knowledge, this is the first demonstration of behavior guided alignment is essential for isolating latent dynamical systems that generalizes across both sessions and animals.
>
> In summary, the LDS module is essential for our scientific goal of testing shared dynamical motifs across individuals. In the revised manuscript, we will add these clarifications to directly address this concern. We welcome further questions and look forward to your reassessment.
>
> > Q2: Will the wheel turning data be made public upon publication?
>
> Yes! We are committed to open science and will release the striatal 2p calcium imaging data upon acceptance. We hope you will consider this as an additional crucial contribution of the paper to the community.
>
> > Q3: Fig. 4a and 4b … for easy comparison.
>
> Great suggestions! For easy comparison, we will put the first column of Fig. 3B to Fig. 4A and the top right similarity matrix in Fig. 3C to Fig. 4B. Then we will move the C-E to the third row in Fig. 4.
>
> > Limitations
>
> Training a session-specific encoder is not a limitation but a necessary design choice due to the substantial heterogeneity in neural recordings across sessions. Behavior signals serve as a key for aligning latent neural dynamics across sessions. Without them, embeddings drift arbitrarily in latent space. As shown in Fig. 4, when both contrastive and behavioral losses are removed, thus removing behavioral supervision, the embeddings fail to align, resulting in failure to recover consistent underlying dynamics. Existing *post hoc* alignment methods similarly depend on behavioral information to align latent embeddings across sessions [1,3], further underscoring the necessity of behavioral signals. Moreover, adapting to new sessions with a new encoder, as demonstrated in Figures 5 and S5, only requires a few trials of data.
>
>
> We hope our clarifications on the methodological, conceptual, biological, and experimental novelties, along with the added content in the Related work and Discussion sections, and the explanation of the usefulness of LDS address your concerns. If there are any other points we can clarify or further experiments we can conduct to strengthen your assessment, please let us know.
>
> [1] M. Safaie, et al. "Preserved neural dynamics across animals performing similar behaviour." Nature 623.7988 (2023): 765-771.
> [2] S. Schneider, Steffen, J. H. Lee, and M. W. Mackenzie. "Learnable latent embeddings for joint behavioural and neural analysis." Nature 617.7960 (2023): 360-368.
> [3] A. Gosztolai, et al. "MARBLE: interpretable representations of neural population dynamics using geometric deep learning." Nature Methods (2025): 1-9.
> [4] Z. Guo, et al. "BLEND: Behavior-guided Neural Population Dynamics Modeling via Privileged Knowledge Distillation." The International Conference on Learning Representations (2025).

---

> > ### Comment · Reviewer_6wRv · 2025-08-04
> >
> > I thank the authors for their response. I believe the proposed changes to the related work section will improve the paper. I also think the data contribution should be very useful for future research into neural decoding and representation learning. I will focus on my primary concern with the paper, the LDS.
> >
> > ## Does the LDS help?
> > I have responded to the main points in the rebuttal's discussion of the LDS below. I have also summarized my current understanding of the evidence at the end.
> >
> > **Can we extract a preserved and latent dynamical system ... underlying neural activity across animals performing similar tasks? Do "shared dynamical motifs" exist across subjects?**
> > I agree that these are both interesting and important scientific questions. However, I don't think the paper in its current form answers these questions.
> >
> > **Figure 5, S5: pretrained and frozen LDS results in high decoding performance**
> > I view this as very indirect evidence that CANDY extracts a preserved dynamical system. One interpretation of Figure 5 is that it simply takes more data to train the LDS well, or that learning the dynamics is a distraction from the decoding task. For example, from eyeballing the two brown lines in Figure 5, it looks like the neural latent dynamics would achieve better results if it simply learned an uninformative LDS (so that the latent dynamics are the same as the latent embeddings). In fact, you would hope that with enough data cross-validation on the behavior decoding would upweight the contrastive and behavior losses relative to the k-step loss so that the dynamics curve would converge to the embeddings curve.
> >
> > **Figures 2, 4: omitting the alignment step prevents recovery of a coherent LDS**
> > I have similar concerns here as with Figure 5. The evidence for the utility of an LDS is very indirect, and an uninformative LDS would be encouraged given enough data. It does look like the contrastive loss could provide a better embedding space to use an LDS with, though, given the continuity of the embeddings in Figure 4A. But this seems to be more an argument for the contrastive loss than the LDS.
> >
> > **Interpretability of the LDS**
> > I agree that the LDS (Eq.s 4-5) is interpretable in the sense that we can calculate the eigenvalues of the dynamics matrix, $A$, to retrieve characteristic damping timescales and oscillatory frequencies. Or we could calculate the spectrum of $\hat{u}$ to help understand the frequency content of the embedding.
> >
> > **My current understanding and questions**
> > I agree that the LDS is a central aspect of the model. If, hypothetically, the LDS didn't add any abilities or interpretability to CANDY, I would consider the paper an interesting application of ranking contrastive learning, bus much less novel and potentially impactful. Currently, I don't think there is good reason to believe that the LDS adds any abilities or interpretability. Here are several ways I think the utility or interpretability benefits of the LDS could be assessed:
> >
> > *Filtering:* One way to demonstrate the utility of the LDS would be to demonstrate improved decoding performance using smoothed or filtered embeddings. However, this isn't observed in the data presented (e.g. Figure 3D vs. E).
> >
> > *Forecasting:* Another positive piece of evidence would be if the LDS allows us to effectively forecast behavior, or even just future embeddings. If this is possible, it would forestall concerns that the LDS is uninformative.
> >
> > *LDS interpretability:* A weaker piece of evidence would just be to check whether the LDS is informative -- are any eigenvalues close to the unit circle? are there prominent oscillations? does the pretrained LDS in Figure 5 display any dynamic characteristics that are apparent in the held out animal's data?

---

> > > ### Author Response · Authors · 2025-08-05
> > >
> > > We appreciate the positive notes provided by the reviewer on the novelty of our work and the importance of our experiments.  We acknowledge reviewer’s expertise in behavioral decoding and close familiarity with prior work such as DFINE [1]. Respectfully, this close interest in behavior may have led to the misunderstanding of our goals and we believe there remain misinterpretations of our core motivation and contributions. Please allow us to elaborate.
> > >
> > > CANDY does not aim to enhance behavioral decoding performance by adding a linear dynamical system (LDS). Rather, our goal is to **introduce a rank-based contrastive learning framework that enforces preservation of behavioral structure within the embedding space, thereby enabling extraction of preserved underlying dynamics**. In this work, the LDS serves as a simple, transparent choice of dynamical model. It helped to illustrate the power of our framework: by aligning neural embeddings across sessions and animals via rank-based contrastive learning, we recover a shared dynamical system (see Figures 2, 3E, and 4). Without LDS, the embeddings would align, but we could not test the existence of a shared dynamical structure.
> > >
> > > In short, our contribution is the development of the first framework for uncovering the shared latent dynamics through contrastive alignment. Reviewer acknowledges this as an important scientific question, though the comments then dive into the follow-up questions on further characterization. We agree this is interesting, but currently out of scope. Note that we are ***not*** claiming that the LDS itself is **superior** or **helpful** for behavior decoding. Regardless, we focus on whether contrastive alignment can reliably extract the preserved dynamics.
> > >
> > > > One interpretation of Figure 5 is that it simply takes more data to train the LDS well
> > >
> > > This speculation is directly refuted in Fig. 4. There, even with plenty of data, an LDS trained *without* our contrastive-alignment term fails to recover a shared dynamical system. This demonstrates that embedding alignment across sessions, enforced by our rank-based contrastive loss, is the crucial ingredient; *simply adding more data is not enough to learn a preserved LDS effectively*.
> > >
> > > > or that learning the dynamics is a distraction from the decoding task
> > >
> > > The impression that learning the dynamics is a distraction from the decoding task is exactly why we believe there is a ***fundamental misunderstanding*** about what our goal is. Our goal is **not** to suggest that learning an LDS boosts decoding performance. Rather, we aim to show that contrastive alignment across sessions is the critical mechanism for uncovering preserved neural dynamics. The LDS is just the vehicle we use to quantify those shared dynamics. Learning the dynamics is not a distraction, but a way to demonstrate that, without proper alignment, the preserved dynamics cannot be revealed (Fig. 2).
> > >
> > > > For example, from eyeballing the two brown lines in Figure 5, it looks like the neural latent dynamics would achieve better results if it simply learned an uninformative LDS (so that the latent dynamics are the same as the latent embeddings)
> > >
> > > We believe there is a misunderstanding of what Fig. 5 is trying to demonstrate: the pretrained temporal structure can be applied to new animals, indicating the existence of such preserved temporal structure. Nevertheless, to put this recurring concern to rest, we explicitly tested the hypothesis that “(CANDY) simply learned an uninformative LDS” by fixing the LDS transition matrix to the identity matrix (A = I, b = 0). Under this uninformative dynamical model, decoding performance drops dramatically on the mouse dataset (entry: average across 5 seeds ± s.e.m.):
> > > |latent dim|CANDY with learnable A| CANDY with uninformative A|
> > > |-|-|-|
> > > |4|0.77±0.02|0.51±0.02|
> > > |8|0.80±0.01|0.54±0.01|
> > >
> > > These results confirm that the LDS must learn meaningful dynamics from neural data since an identity model is clearly insufficient. Thereby refuting the hypothesis that CANDY simply converges to an uninformative, trivial dynamical system. We will include this additional experiment in the Appendix.
> > >
> > > > In fact, you would hope that with enough data cross-validation on the behavior decoding would upweight the contrastive and behavior losses relative to the k-step loss so that the dynamics curve would converge to the embeddings curve.
> > >
> > > This statement is speculative in nature and we have shown it to be incorrect based on our results. First, the contrastive objective is essential for extracting the preserved dynamical system (see Figures 2 and 4). Second, the dynamical system is effectively learned since the decoding performance from CANDY with an uninformative dynamics matrix is substantially lower (see the table above).

---

> > > > ### Author Response · Authors · 2025-08-05
> > > >
> > > > > The evidence for the utility of an LDS is very indirect, and
> > > >
> > > > First, our aim is to demonstrate that with effective *contrastive alignment*, one can recover a *shared dynamical structure* across sessions through LDS. Second, the utility of LDS is directly shown in Fig. 2: we use the learned dynamics matrix A to plot the dynamical flow field, and without it there’s no way to verify that our embedding alignment truly recovers the preserved dynamical system.
> > > >
> > > > > an uninformative LDS would be encouraged given enough data
> > > >
> > > > We explicitly tested this above, which turned out to be not true in our case: fixing the dynamics matrix to an identity matrix ($A = I$) causes decoding accuracy to drop dramatically, confirming that CANDY learns a non-trivial LDS rather than defaulting to an uninformative identity map.
> > > >
> > > > > It does look like the contrastive loss could provide a better embedding space to use an LDS with, though, given the continuity of the embeddings in Figure 4A. But this seems to be more an argument for the contrastive loss than the LDS
> > > >
> > > > Yes, our key contribution lies in demonstrating the power of the contrastive loss for aligning neural embeddings, not in the choice of LDS itself. In Fig. 4, we show that *simply applying a behavior-supervision loss fails to align embeddings across sessions to recover the preserved dynamical system*. In contrast, adding our rank-based contrastive term produces a continuous embedding space from which a shared dynamical system can be reliably inferred. The LDS here serves as a *tool* to quantify and illustrate the extracted shared dynamics; the central innovation is that the contrastive alignment makes this extraction possible.
> > > >
> > > > > the LDS is a central aspect of the model
> > > >
> > > > We apologize if our earlier explanation of LDS gave the wrong impression. As stated before, the main focus and novelty of CANDY is that the contrastive alignment is essential for extracting preserved dynamics across sessions and animals. LDS is a key component in the sense that it can verify if the preserved dynamics is correctly extracted as shown in Fig. 2, but it is not intended to enhance behavior decoding performance. Notably, no prior work has discussed the important role of embedding alignment for extracting preserved dynamics across sessions.
> > > >
> > > > > If, hypothetically, the LDS didn't add any abilities or interpretability to CANDY, I would consider the paper an interesting application of ranking contrastive learning, but much less novel and potentially impactful
> > > >
> > > > We believe our contributions go far beyond just simply using rank-based contrastive learning on neural data. Overall, we emphasize that our contributions include
> > > > 1) Methodologically, we **develop a new framework to extract preserved dynamics** underlying continuous behavior across sessions and animals using a rank-based contrastive objective and a linear dynamical system.
> > > > 2) Conceptually, we show that **contrastive alignment is essential for uncovering preserved dynamical systems**, not merely a decoding aid.
> > > > 3) Biologically, while prior work shows preserved neural motifs across animals [2], none have tested if these can arise from the same dynamical system. We show that a dynamical model trained on one set of animals transfers to held-out individuals, **indicating the existence of preserved dynamical structures in a specific brain region under a given task**.
> > > > 4) Experimentally, **we collected a new cross-day multi-animal 2p calcium dataset and will release it if accepted**. ”
> > > >
> > > > We will revise the contributions at the end of the introduction accordingly.
> > > >
> > > > > final experimental suggestions
> > > >
> > > > To address your concerns and show LDS adds abilities, although not the primary focus of our work, we added the following behavior forecasting experiment as proposed (entry: average mean square error across 8 sessions ± s.e.m.):
> > > >
> > > > Mouse dataset
> > > > |forecasting step|t=0|t=1|t=2|t=3|t=4|
> > > > |-|-|-|-|-|-|
> > > > |CANDY|0.0287±0.0035|0.0354±0.0068|0.0415±0.0088|0.0587±0.0089|0.2476±0.0238|
> > > > |DFINE|0.0886±0.0059|0.0930±0.0084|0.1002±0.0092|0.1420±0.0108|0.3882±0.0304|
> > > >
> > > > Monkey dataset
> > > > |forecasting step|t=0|t=1|t=2|
> > > > |-|-|-|-|
> > > > |CANDY|0.0420±0.0035|0.0576±0.0026|0.0766±0.0027|
> > > > |DFINE|0.2635±0.0045|0.2675±0.0055|0.3008±0.0065|
> > > >
> > > > We used a universal behavior decoder across all sessions for CANDY and session-specific behavior decoder for DFINE. This result shows that CANDY has forecasting capability thus the learned dynamical system is not uninformative. We will add this to the Appendix.
> > > >
> > > > [1] Abbaspourazad, Hamidreza, et al. "Dynamical flexible inference of nonlinear latent factors and structures in neural population activity." Nature Biomedical Engineering 8.1 (2024): 85-108.
> > > > [2] Safaie, Mostafa, et al. "Preserved neural dynamics across animals performing similar behaviour." Nature 623.7988 (2023): 765-771.

---

> > > > > ### Author Response · Authors · 2025-08-05
> > > > >
> > > > > We hope our clarifications on the central focus of the paper and the additional experiments on showing that the LDS is informative (fixing A to be identity and forecasting) would address your concerns. If there are any other points we can clarify to strengthen your assessment, please let us know. We look forward to your reassessment!

---

> > > > > > ### Comment · Reviewer_6wRv · 2025-08-08
> > > > > >
> > > > > > Apologies for the delayed response.
> > > > > >
> > > > > > I respectfully disagree that I have a fundamental misunderstanding about the goals of the paper. The paper sets out to extract behavior-relevant preserved neural dynamics across sessions and subjects. When I asked "Does the LDS help?", I intended it in a broader sense than "Does the LDS help behavior decoding?", closer to "Does the LDS help the aims of the paper?" I apologize if this was confusing.
> > > > > >
> > > > > > Perhaps contributing to this confusion, the results that highlight the use the LDS do so through  performance on behavior decoding tasks (Figures 3E, 4E, 5). By no means do I insist that the LDS improve behavior decoding. Although improved behavior decoding would be interesting and useful, there are other worthy goals that the LDS could fill, such as aiding interpretability, enabling us to test the existence of a shared dynamical structure, or to extract preserved neural dynamics. I responded to claims of extracting shared dynamics of neural data using the provided evidence, which is conveyed through the use of behavior decoding $R^2$ values (Figures 3E, 4E, 5).
> > > > > >
> > > > > > **New Forecasting Results**
> > > > > > The new forecasting results show slowly increasing MSEs as the number of forecasted timesteps increases, good evidence that the LDS has successfully modeled the underlying behavior-relevant neural dynamics. Converting the MSEs to $R^2$ values would help the reader interpret how well these dynamics are modeled. I applaud the authors for including this validation and will raise my score accordingly.
> > > > > >
> > > > > > **My Primary Remaining LDS Concern**
> > > > > > Thank you for bringing up Figure 2, which shows the filtered $z$'s along with the LDS vector field, showing alignment of dynamics across the three "animals." This is a very clear demonstration of finding shared dynamics. **The LDS accurately describes the latent dynamics of all three animals.**
> > > > > >
> > > > > > **I would like to know whether the LDS accurately describes the latent dynamics across subjects and sessions for real data.** To sum up my dissatisfaction with the LDS in the original manuscript: how should a practitioner conclude that shared neural dynamics have been found? Simply fitting a model with an LDS to data from multiple individuals should not guarantee shared neural dynamics will be found. A demonstration that the LDS accurately describes the neural dynamics across the subjects, even if it is just 2-dimensional as in Figure 2, would fill this gap and greatly improve the paper, and also provide useful guidance to future users of CANDY.
> > > > > >
> > > > > > Of course, it is very late in the reviewer/author discussion period, so I don't expect this to be completed. But if CANDY is published, please consider it for the camera-ready version.

---

> > > > > > > ### Comment · Reviewer_6wRv · 2025-08-08
> > > > > > >
> > > > > > > **Secondary Points and Responses**
> > > > > > >
> > > > > > > * I agree that the contrastive loss term encourages a "smoother" latent space that aligns different sessions, likely making an LDS more effective at modeling the embedding dynamics. This has been observed in papers such as [3] and [4], albeit with different contrastive setups.
> > > > > > >
> > > > > > > * *Figure 5, not enough data:* The LDS and behavior decoder in the blue lines in Figure 5 have seen data from 2 mice, while the brown lines have seen data from only 1 mouse. The blue lines have seen more data and show higher performance, so I would not interpret the results as a confirmation of preserved temporal structure across subjects.
> > > > > > >
> > > > > > > * *Figure 5b would achieve better results with uninformative LDS* I believe my point here was slightly misinterpreted. If $A = 0$, $Q=\frac{1}{\epsilon}I$, $C=I$, and $R=\epsilon I$ for $0< \epsilon \ll 1$, then the filtered embedding is the per-timestep embedding. If this were the case, Figures 5A and B would show identical results. Since the filtered embedding bpredictions in 5B are worse than the per-timestep predictions in 5A, an optimization of the filtered embedding decoding performance (for example, via cross validation) could in principle discover the above uninformative LDS and improve its performance. I did not intend this a suggested experiment, only to make the point that optimizing decoding performance through cross-validation could, in theory, discourage informative dynamics, as an example of another difficulty in interpreting Figure 5 as evidence of preserved dynamics.
> > > > > > >
> > > > > > >
> > > > > > > > In short, our contribution is the development of the first framework for uncovering the shared latent dynamics through contrastive alignment.
> > > > > > >
> > > > > > > * I believe this is an overstatement. I know of two works that combine multimodal and dynamical system learning contrastive setup, although not in a neuroscience setting [1,2].
> > > > > > >
> > > > > > > > Notably, no prior work has discussed the important role of embedding alignment for extracting preserved dynamics across sessions.
> > > > > > >
> > > > > > > * I also believe this is well appreciated by the wider contrastive learning community. For example, see the discussion of learning only shared information in [2].
> > > > > > >
> > > > > > > [1] Sun, Yuchong, et al. "Long-form video-language pre-training with multimodal temporal contrastive learning." *Advances in neural information processing systems* 35 (2022): 38032-38045.
> > > > > > > [2] Liu, Shengzhong, et al. "Focal: Contrastive learning for multimodal time-series sensing signals in factorized orthogonal latent space." *Advances in Neural Information Processing Systems* 36 (2023): 47309-47338.
> > > > > > > [3] Eysenbach, Benjamin, et al. "Inference via interpolation: Contrastive representations provably enable planning and inference." *Advances in Neural Information Processing Systems* 37 (2024): 58901-58928.
> > > > > > > [4] Wang, Tongzhou, and Phillip Isola. "Understanding contrastive representation learning through alignment and uniformity on the hypersphere." *International conference on machine learning*. PMLR, 2020.

---

> > > > > > > > ### Author Response · Authors · 2025-08-09
> > > > > > > >
> > > > > > > > We appreciate your positive notes on the LDS experiments and are glad that the original Fig. 2 and the new forecasting analyses help clarify the LDS’s utility. We apologize for the previous misinterpretation of your “does LDS help” question, and your clarification on your questions and arguments was very helpful! We will update the manuscript as suggested and include citations of the broader field, not just limited to neuroscience, in the Related work.
> > > > > > > >
> > > > > > > > Based on your suggestions, we will update the manuscript to emphasize the forecasting performance in order to better illustrate that the learned dynamics are informative.
> > > > > > > >
> > > > > > > > To support the claim in Fig. 5 regarding the generalization of the pretrained LDS to a new animal, we conducted an additional synthetic experiment. In this setting, we held out one session that shared the same spiral latent dynamics but had a different nonlinear observation readout and noise statistics. By fixing the LDS and updating only the encoder for the held-out session, the model successfully recovered the spiral latent trajectory, closely following the pretrained LDS flow field. We will include this visualization for the held-out synthetic session in the revised manuscript.
> > > > > > > >
> > > > > > > > To address the reviewer’s concern regarding the real data, we will show the flow field by projecting the D=4 dynamics into a 2D state space using principal component analysis in the revised manuscript. We observed that the latent dynamical factors $z$ follow the flow field and are aligned across sessions. Quantitatively, we will show the forecasting capability of the latent dynamical factors as a function of the forecasting time step.
> > > > > > > >
> > > > > > > > We also appreciate the references you provided on contrastive learning for discovering latent temporal structure in other domains. To avoid overstatement and position our work better within this broader literature beyond neuroscience, we will add the following sentence at L78: “Furthermore, contrastive learning has been shown to be effective for extracting shared temporal structure from time-series data across modalities in other fields, such as videos and sensing physics [1–2].”
> > > > > > > >
> > > > > > > > We sincerely thank the reviewer for the engagement with our work and for the updated score. Your suggestions have helped us strengthen both the conceptual framing and empirical validation of our work. We truly appreciate your support and the time and care you have invested in the review process.
> > > > > > > >
> > > > > > > > [1] Sun, Yuchong, et al. "Long-form video-language pre-training with multimodal temporal contrastive learning." Advances in neural information processing systems 35 (2022): 38032-38045. [2] Liu, Shengzhong, et al. "Focal: Contrastive learning for multimodal time-series sensing signals in factorized orthogonal latent space." Advances in Neural Information Processing Systems 36 (2023): 47309-47338.

---

### Note · Authors · 2025-08-12

We sincerely thank AC for handling our submission and all reviewers for their thoughtful reviews and the encouraging words on the potential impact of this work in neuroscience.

**Key contributions of our work**: The central novelty lies in the following four aspects, rather than improving decoding performance: **methodologically**, we propose a new framework, CANDY, that uses a ranked-based contrastive objective and a linear dynamical system (LDS) to extract interpretable and preserved dynamics across sessions and animals; **conceptually**, we demonstrate the effectiveness of contrastive alignment for uncovering preserved dynamics; **biologically**, we provide evidence for the existence of preserved dynamics in certain brain regions under a given task; and **experimentally**, we collected a new cross-day 2p calcium dataset from the mouse striatum (3 animals, 15 sessions). We validate our framework on a synthetic dataset (Fig. 2), a mouse calcium imaging dataset (Figs. 3–5, S1–S2), and a monkey electrophysiology dataset (Figs. S3–S5), demonstrating that CANDY generalizes across species and recording modalities.

**Efforts to address reviewers’ comments**: In response to the feedback from Reviewers 6wRv, oySm and JGvB, we expanded the Related work with suggested references and Discussion to include nonlinear behavior readouts, identification of different latent dynamical systems among animals, and generalization to unseen trial types. We also clarified the Methods section following Reviewer oySm’s feedback. Furthermore, we included an additional baseline model which uses a shared dynamical system as suggested by Reviewer P5Sz. Finally, based on the suggestions from Reviewer 6wRv and JGvB, we strengthened the evidence for the utility and interpretability of the LDS by referring to Fig. 2 and including the forecasting result and a figure of 2D projected flow field.

We will release the code and the new cross-day 2p striatum dataset upon publication. We believe these clarifications and revisions address all major concerns and significantly strengthen the paper’s clarity, reproducibility, and value to the neuroscience community.

We especially appreciate the active engagement from Reviewers 6wRv, oySm, and JGvB during the discussion period. However, we’d also like to express our concern that Reviewer P5Sz did not participate in the discussion period, yet still clicked the Mandatory Acknowledgement, which appears inconsistent with the public discussion record.

---

### Decision · Program_Chairs · 2025-09-17

**Decision:**

Accept (spotlight)

**Comment:**

The authors propose a poweful new framework for learning cross-animal shared dynamics from neural recordings called CANDY using a novel contrastive objective. This is an exciting step away from simple representation learning to latent dynamics alignment using contrastive learning with a potential to be useful for "understanding". The experiments and comparisons were satisfactory, especially the neuroscience datasets. The authors rebuttal was great. Nice work.

Although completely different in methodology and approach, the AC believes that the following paper attempts to solve a generalized version of the problem. This is just for information. No action needed.
Vermani, A., Nassar, J., Jeon, H., Dowling, M., & Park, I. M. (2024, October 7). Meta-dynamical state space models for integrative neural data analysis. The Thirteenth International Conference on Learning Representations (ICLR). https://openreview.net/forum?id=SRpq5OBpED